# Accelerating Visual-Policy Learning through Parallel Differentiable Simulation

**Haoxiang You**
Department of Mechanical Engineering
Yale University
New Haven, CT 06520
`haoxiang.you@yale.edu`

**Yilang Liu**
Department of Mechanical Engineering
Yale University
New Haven, CT 06520
`yilang.liu@yale.edu`

**Ian Abraham**
Department of Mechanical Engineering
Department of Computer Science
Yale University
New Haven, CT 06520
`ian.abraham@yale.edu`

## Abstract

In this work, we propose a computationally efficient algorithm for visual policy learning that leverages differentiable simulation and first-order analytical policy gradients. Our approach decouple the rendering process from the computation graph, enabling seamless integration with existing differentiable simulation ecosystems without the need for specialized differentiable rendering software. This decoupling not only reduces computational and memory overhead but also effectively attenuates the policy gradient norm, leading to more stable and smoother optimization. We evaluate our method on standard visual control benchmarks using modern GPU-accelerated simulation. Experiments show that our approach significantly reduces wall-clock training time and consistently outperforms all baseline methods in terms of final returns. Notably, on complex tasks such as humanoid locomotion, our method achieves a $4\times$ improvement in final return, and successfully learns a humanoid running policy within 4 hours on a single GPU. Videos and code are available on `https://haoxiangyou.github.io/Dva_website/`

## 1   Introduction

Learning to control robots from visual inputs is a key challenge in robotics, with the potential to enable a wide range of real-world applications, ranging from autonomous driving and home service robots to industrial automation. Most methods for learning visual policies fall into two categories: imitation learning and reinforcement learning (RL). Imitation learning trains policies by mimicking expert demonstrations, which are typically collected via human operation [Bojarski et al., 2016, Kendall et al., 2019] or teleoperation systems [Chi et al., 2024, Black et al., 2024]. When expert demonstrations are scarce or difficult to obtain, visual policies can instead be learned through RL [Mnih et al., 2015, Hafner et al., 2019]. However, RL methods typically require long training times and substantial computational resources, such as large-scale GPU clusters, to achieve effective control.

Recent advances in differentiable simulation have enabled alternative policy optimization methods, known as analytical policy gradients (APG) [Freeman et al., 2021, Xu et al., 2021, Schwarke et al., 2024]. These methods achieve much higher computational efficiency by replacing zeroth-order

39th Conference on Neural Information Processing Systems (NeurIPS 2025).

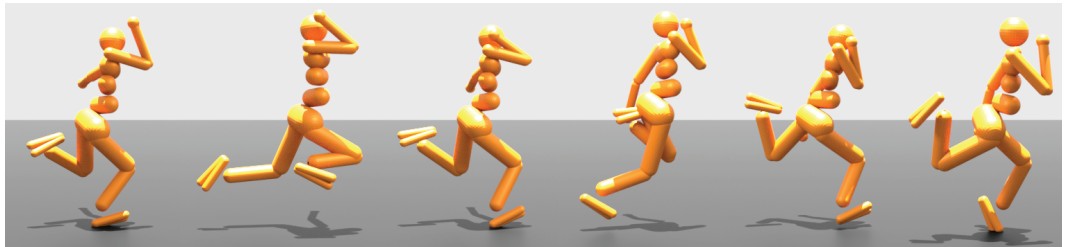

Figure 1: **Fast training of humanoid running policy from pixel input.** Our method learns a stable running gait in 4 hours on a single RTX 4080 GPU.

gradient estimates with first-order gradients. However, extending APG methods to visual control remains challenging: obtaining high-quality differentiable rendering is non-trivial, and computing Jacobians over pixel inputs is both memory- and computationally inefficient. Consequently, existing approaches either train separate differentiable renderers [Wiedemann et al., 2023, Liu et al., 2024] or rely on low-dimensional visual features, which demand considerable engineering effort [Heeg et al., 2024, Luo et al., 2024].

In this work, we propose Decoupled Visual-Based Analytical Policy Gradient (D.Va), a novel method for learning visual policies using differentiable simulation. The core idea is to decouple visual observations from the computation graph, eliminating the need to differentiate through the rendering process. We find that this decoupling not only improves memory and computational efficiency by avoiding Jacobian computations over pixel space, but also normalizes the policy gradient, making visual policy learning more stable. We further provide a formal analysis of our new computation graph, demonstrating that the proposed decoupling policy gradient can be interpreted as a form of policy distillation from open-loop trajectory optimization. This reveals a fundamental connection between open-loop trajectory optimization and closed-loop policy learning.

Finally, we benchmark a diverse set of visual policy learning methods using a GPU-accelerated simulation platform that supports parallelized physics and rendering, providing a robust and scalable testbed for evaluation. Our comparisons include the proposed D.Va, two model-free RL algorithms [Laskin et al., 2020a, Yarats et al., 2021], the model-based RL method DreamerV3 [Hafner et al., 2023], an analytical policy gradient method with differentiable rendering, and state-to-visual distillation [Mu et al., 2025]—a two-stage framework that first trains a state-based policy and then distills it into a visual policy through imitation learning. Experiments highlight D.Va's superior computational efficiency across a wide range of control tasks.

In summary, our contributions are: (a) Proposing D.Va, a computationally efficient method for visual policy learning; (b) Benchmarking diverse visual policy learning approaches on state-of-the-art simulation platforms; (c) Providing an analysis of analytical policy gradients and highlighting the new opportunities for integrating policy learning with trajectory optimization techniques.

## 2 Background

In this section, we formally define the policy optimization problem and introduce key concepts in analytical policy gradient methods.

### 2.1 Problem formulation

We consider the dynamical system $\mathbf{s}_{t+1} = f(\mathbf{s}_t, \mathbf{a}_t)$, where $\mathbf{s}_t \in \mathcal{S}$ denotes the state and $\mathbf{a}_t \in \mathcal{A}$ the action. The dynamics function $f : \mathcal{S} \times \mathcal{A} \to \mathcal{S}$ is assumed to be fully differentiable with respect to both state and action. Let $\mathbf{o}_t = g(\mathbf{s}_t) \in \mathcal{O}$ denote the observation, where $g : \mathcal{S} \to \mathcal{O}$ is the sensor model. Throughout this paper, we assume $\mathbf{s}_t$ is a low-dimensional internal representation of the robot (e.g., joint states), while $\mathbf{o}_t$ represents observation derived from $\mathbf{s}_t$ (e.g., images).

Consider the trajectory $\tau = \{\mathbf{s}_0, \mathbf{a}_0, \mathbf{s}_1, \mathbf{a}_1, \ldots \mathbf{s}_T, \mathbf{a}_T\}$, which is a sequence of state-action pairs with horizon $T$. The total return is defined as $\mathcal{J}(\tau) = \sum_{i=0}^{T} \gamma^t R(\mathbf{s}_t, \mathbf{a}_t)$, where $\gamma \in (0, 1)$ is the discount factor, and $R : \mathcal{S} \times \mathcal{A} \to \mathbb{R}$ is the reward function. We denotes the discounted temporal reward as $r_t = \gamma^t R(\mathbf{s}_t, \mathbf{a}_t)$. A feedback policy $\pi(\cdot | \mathbf{o}_t, \boldsymbol{\theta}) : \mathcal{O} \times \boldsymbol{\Theta} \to \Delta(\mathcal{A})$ is a family of conditional probability distributions that maps an observation to a probability distribution over

actions. Typically, this distribution is modeled as a Gaussian, allowing the action to be expressed via the reparameterization trick: $\mathbf{a}_t = \boldsymbol{\mu}(\mathbf{o}_t, \boldsymbol{\theta}) + \boldsymbol{\sigma}(\mathbf{o}_t, \boldsymbol{\theta}) \odot \boldsymbol{\epsilon}_t$ where $\boldsymbol{\mu} : \mathcal{O} \times \Theta \to \mathcal{A}$, $\boldsymbol{\sigma} : \mathcal{O} \times \Theta \to \mathcal{A}$ represent the mean and standard deviation respectively, and $\boldsymbol{\epsilon}_t \sim \mathcal{N}(\mathbf{0}, \mathbf{I})$ is injected noise. Given an initial condition $\mathbf{s}_0$, and a sequence of injected noises $\mathcal{E} = \{\boldsymbol{\epsilon}_0, \boldsymbol{\epsilon}_1, \dots \boldsymbol{\epsilon}_T\}$, a trajectory $\tau$ can be generated by rolling out from policy under dynamics $f$ and sensor model $g$, which we explicitly denoted as $\tau(\mathbf{s}_0, \boldsymbol{\theta}, \mathcal{E})$ and the corresponding return as $\mathcal{J}(\tau(\mathbf{s}_0, \boldsymbol{\theta}, \mathcal{E}))$. For notational simplicity, we omit the explicit trajectory $\tau$ and write the return directly as $\mathcal{J}(\mathbf{s}_0, \boldsymbol{\theta}, \mathcal{E})$. The expected return for a given policy is defined as $\mathcal{V}(\boldsymbol{\theta}) = \mathbb{E}_{\mathbf{s}_0 \sim \rho_0} \mathbb{E}_{\boldsymbol{\epsilon}_t \overset{\text{i.i.d.}}{\sim} \mathcal{N}(\mathbf{0}, \mathbf{I})} \mathcal{J}(\mathbf{s}_0, \boldsymbol{\theta}, \mathcal{E})$, where $\rho_0$ is initial distribution. The goal of policy optimization is to find policy parameters $\boldsymbol{\theta}$ maximizing the expected return.

## 2.2 Analytical policy gradient

Here, we provide background on analytical policy gradient (APG) methods. These methods compute the policy gradient as

$$\nabla_{\boldsymbol{\theta}} \mathcal{V} = \mathbb{E}_{\mathbf{s}_0 \sim \rho_0} \mathbb{E}_{\boldsymbol{\epsilon}_t \overset{\text{i.i.d.}}{\sim} \mathcal{N}(\mathbf{0}, \mathbf{I})} \nabla_{\boldsymbol{\theta}} \mathcal{J}(\mathbf{s}_0, \boldsymbol{\theta}, \mathcal{E}) = \mathbb{E}_{\mathbf{s}_0 \sim \rho_0} \mathbb{E}_{\boldsymbol{\epsilon}_t \overset{\text{i.i.d.}}{\sim} \mathcal{N}(\mathbf{0}, \mathbf{I})} \left[ \sum_{t=0}^{T} \nabla_{\boldsymbol{\theta}} r_t \right], \tag{1}$$

where the gradient of each term in the sum is given by

$$\nabla_{\boldsymbol{\theta}} r_t = \frac{\partial r_t}{\partial \mathbf{a}_t} \frac{d\mathbf{a}_t}{d\boldsymbol{\theta}} + \frac{\partial r_t}{\partial \mathbf{s}_t} \frac{d\mathbf{s}_t}{d\boldsymbol{\theta}}, \ t = 0, \dots, T$$

$$\frac{d\mathbf{a}_t}{d\boldsymbol{\theta}} = \frac{\partial \mathbf{a}_t}{\partial \boldsymbol{\theta}} + \frac{\partial \mathbf{a}_t}{\partial \mathbf{o}_t} \frac{d\mathbf{o}_t}{d\mathbf{s}_t} \frac{d\mathbf{s}_t}{d\boldsymbol{\theta}}, \ t = 0, \dots, T$$

$$\frac{d\mathbf{s}_t}{d\boldsymbol{\theta}} = \frac{\partial \mathbf{s}_t}{\partial \mathbf{s}_{t-1}} \frac{d\mathbf{s}_{t-1}}{d\boldsymbol{\theta}} + \frac{\partial \mathbf{s}_t}{\partial \mathbf{a}_{t-1}} \frac{d\mathbf{a}_{t-1}}{d\boldsymbol{\theta}}, \ t = 1 \dots T \text{ and } \frac{d\mathbf{s}_0}{d\boldsymbol{\theta}} = \mathbf{0}. \tag{2}$$

Here, $\frac{d}{d}$ denotes the total derivative, and $\frac{\partial}{\partial}$ represents the partial derivative, both expressed in matrix form as Jacobians. The expectation can then be estimated via empirical sum. As these methods estimate the first-order policy gradient by backpropagation through trajectories, they are also named as first-order policy gradients (FoPG) or backpropagation through time (BPTT).

**Short-horizon actor critic (SHAC)** The trajectory gradient, i.e., $\nabla_{\boldsymbol{\theta}} \mathcal{J}(\mathbf{s}_0, \boldsymbol{\theta}, \mathcal{E})$, can quickly become intractable as the horizon $T$ increases due to exploding gradients by multiplying a series of matrices. The exploding gradient of an individual trajectory leads to the high-variance empirical estimate of the policy gradient [Metz et al., 2021], as well as an empirical bias problem [Suh et al., 2022]. Fortunately, these problems are largely solved by the Short-Horizon Actor-Critic (SHAC) method [Xu et al., 2021]. The key idea is to truncate the long trajectory into smaller segments and incorporate a learned value function for long-horizon predictions. More specifically, at each iteration, the SHAC algorithm optimizes the following actor loss

$$\mathcal{L}_{\boldsymbol{\theta}} = -\frac{1}{Nh} \sum_{i=1}^{N} \left[ \left( \sum_{t=t_0}^{t_0+h-1} \gamma^{t-t_0} R(\mathbf{s}_t^{(i)}, \mathbf{a}_t^{(i)}) \right) + \gamma^h V_{\phi}(\mathbf{s}_{t_0+h}^{(i)}) \right], \tag{3}$$

where $\mathbf{s}_t^{(i)}$ and $\mathbf{a}_t^{(i)}$ are states and actions of the $i$-th trajectory rollout, and $V_{\phi} : \mathcal{S} \to \mathbb{R}$ is the value function learned with TD-$\lambda$ tricks [Sutton et al., 1998].

## 3 Method

In this work, we extend SHAC to handle complex observations, e.g., images, whereas the original SHAC primarily focused on low-dimensional state spaces. The main challenge in visual setting is calculating the observation Jacobian $\frac{d\mathbf{o}_t}{d\mathbf{s}_t}$, which is both computationally and memory intensive due to the high dimensionality of images. To address this, we omit all terms of the form $\left( \frac{\partial \mathbf{a}_t}{\partial \mathbf{o}_t} \frac{d\mathbf{o}_t}{d\mathbf{s}_t} \frac{d\mathbf{s}_t}{d\boldsymbol{\theta}} \right)$, resulting in a quasi-policy gradient that we refer to as the decoupled policy gradient (DPG). In this section, we provide a formal analysis showing how the DPG works, as well as algorithm that utilizes DPG for training a visual policy.

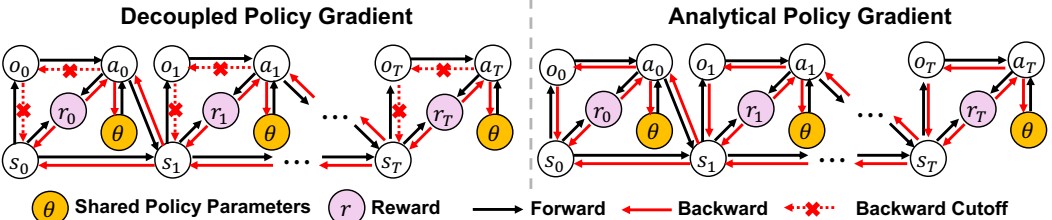

Figure 2: **Computation graphs of decoupled policy gradient (DPG) and APG.** The policy gradient is traced from any reward $r_t$, propagated backward through the graph to the shared parameters $\boldsymbol{\theta}$. APG backpropagates through the entire pipeline, whereas the DPG prevents gradient flow through the rendering process.

## 3.1 Decoupled policy gradient

We divide the analytical policy gradient (1) into two parts by separating all terms involving $\frac{d\mathbf{o}_t}{d\mathbf{s}_t}$ from those that do not: $\nabla_{\boldsymbol{\theta}}\mathcal{V} = \tilde{\nabla}_{\boldsymbol{\theta}}\mathcal{V} + \mathcal{B}$, where

$$\tilde{\nabla}_{\boldsymbol{\theta}}\mathcal{V} = \mathbb{E}_{\mathbf{s}_0 \sim \rho_0}\mathbb{E}_{\boldsymbol{\epsilon}_t \overset{\text{i.i.d.}}{\sim} \mathcal{N}(\mathbf{0},\mathbf{I})}\Big[\tilde{\nabla}_{\boldsymbol{\theta}}\mathcal{J}(\mathbf{s}_0,\boldsymbol{\theta},\mathcal{E})\Big] = \mathbb{E}_{\mathbf{s}_0 \sim \rho_0}\mathbb{E}_{\boldsymbol{\epsilon}_t \overset{\text{i.i.d.}}{\sim} \mathcal{N}(\mathbf{0},\mathbf{I})}\Big[\sum_{t=0}^{T}\tilde{\nabla}_{\boldsymbol{\theta}}r_t\Big]$$

$$\tilde{\nabla}_{\boldsymbol{\theta}}r_t = \frac{\partial r_t}{\partial \mathbf{a}_t}\frac{\partial \mathbf{a}_t}{\partial \boldsymbol{\theta}} + \frac{\partial r_t}{\partial \mathbf{s}_t}\frac{d\tilde{\mathbf{s}}_t}{d\boldsymbol{\theta}}, \ t = 0 \dots T$$

$$\frac{d\tilde{\mathbf{s}}_t}{d\boldsymbol{\theta}} = \frac{\partial \mathbf{s}_t}{\partial \mathbf{s}_{t-1}}\frac{d\tilde{\mathbf{s}}_{t-1}}{d\boldsymbol{\theta}} + \frac{\partial \mathbf{s}_t}{\partial \mathbf{a}_{t-1}}\frac{\partial \mathbf{a}_{t-1}}{\partial \boldsymbol{\theta}}, \ t = 1 \dots T \text{ and } \frac{d\tilde{\mathbf{s}}_0}{d\boldsymbol{\theta}} = \mathbf{0}, \quad (4)$$

and

$$\mathcal{B} = \mathbb{E}_{\mathbf{s}_0 \sim \rho_0}\mathbb{E}_{\boldsymbol{\epsilon}_t \overset{\text{i.i.d.}}{\sim} \mathcal{N}(\mathbf{0},\mathbf{I})}\Big[\sum_{t=0}^{T}\Big(\frac{\partial r_t}{\partial \mathbf{a}_t}\frac{\partial \mathbf{a}_t}{\partial \mathbf{o}_t}\frac{d\mathbf{o}_t}{d\mathbf{s}_t}\frac{d\mathbf{s}_t}{d\boldsymbol{\theta}}\Big)\Big]$$

$$\frac{d\mathbf{s}_t}{d\boldsymbol{\theta}} = \frac{\partial \mathbf{s}_t}{\partial \mathbf{a}_{t-1}}\Big(\frac{\partial \mathbf{a}_{t-1}}{\partial \boldsymbol{\theta}} + \frac{\partial \mathbf{a}_{t-1}}{\partial \mathbf{o}_{t-1}}\frac{d\mathbf{o}_{t-1}}{d\mathbf{s}_{t-1}}\frac{d\mathbf{s}_{t-1}}{d\boldsymbol{\theta}}\Big) + \frac{\partial \mathbf{s}_t}{\partial \mathbf{s}_{t-1}}\frac{d\mathbf{s}_{t-1}}{d\boldsymbol{\theta}}, \ t = 1 \dots T \text{ and } \frac{d\mathbf{s}_0}{d\boldsymbol{\theta}} = \mathbf{0}. \quad (5)$$

Detailed derivation of this decomposition is provided in Appendix A. We refer to $\tilde{\nabla}_{\boldsymbol{\theta}}\mathcal{V}$ (4) as the *decoupled policy gradient*, which improves the policy by distilling results from open-loop trajectory optimizations. We denote $\mathcal{B}$ (5) as *control regularization* which captures the interdependence between actions. Below, we provide a conceptual explanation of each term, describe the rationale behind their naming, and validate the effectiveness of the decoupled policy gradient through experiments.

**Distilling from open-loop Trajectories** Here, we show how the decoupled policy gradient (4) can be interpreted as distilling from open-loop trajectories. We begin by initializing a open-loop sequence of controls, $\mathbf{A} = \{\mathbf{a}_0, \mathbf{a}_1, \dots, \mathbf{a}_T\}$, by rolling out the policy $\pi(\cdot|\cdot;\boldsymbol{\theta})$ under the initial condition $\mathbf{s}_0$. Given the open-loop control sequence $\mathbf{A}$ and initial condition $\mathbf{s}_0$, the sequence of states $\mathbf{S} = \{\mathbf{s}_0, \mathbf{s}_1, \dots, \mathbf{s}_T\}$ and observation $\mathbf{O} = \{\mathbf{o}_0, \mathbf{o}_1, \dots, \mathbf{o}_T\}$ can be reconstructed via dynamics $f$ and sensor model $g$. In this case, the return $\mathcal{J}(\mathbf{s}_0, \mathbf{A}) = \sum_{t=0}^{T} r_t$ is solely a function of the initial condition $\mathbf{s}_0$ and the control sequence $\mathbf{A}$. The gradient of return with respect to the control sequence $\mathbf{A}$ is given by

$$\nabla_{\mathbf{A}}\mathcal{J} = \{\nabla_{\mathbf{a}_0}\mathcal{J}, \nabla_{\mathbf{a}_1}\mathcal{J}, \dots, \nabla_{\mathbf{a}_T}\mathcal{J}\}, \text{where } \nabla_{\mathbf{a}_t}\mathcal{J} = \sum_{j=t}^{T}\nabla_{\mathbf{a}_t}r_j, \text{ and}$$

$$\nabla_{\mathbf{a}_t}r_j = \begin{cases} \frac{\partial r_j}{\partial \mathbf{s}_j}\frac{\partial \mathbf{s}_j}{\partial \mathbf{s}_{j-1}}\frac{\partial \mathbf{s}_{j-1}}{\partial \mathbf{s}_{j-2}}\cdots\frac{\partial \mathbf{s}_{t+2}}{\partial \mathbf{s}_{t+1}}\frac{\partial \mathbf{s}_{t+1}}{\partial \mathbf{a}_t}, \text{when } j > t \\ \frac{\partial r_t}{\partial \mathbf{a}_t}, \text{when } j = t \end{cases}, \ t = 0, \dots T, \ j = t, \dots T. \quad (6)$$

We improve the control sequence by taking a small step $\beta$ in the gradient direction for each action

$$\bar{\mathbf{a}}_t = \mathbf{a}_t + \beta\nabla_{\mathbf{a}_t}\mathcal{J}. \quad (7)$$

We denote the updated sequence as $\bar{\mathbf{A}} := \{\bar{\mathbf{a}}_0, \bar{\mathbf{a}}_1, \dots, \bar{\mathbf{a}}_T\}$. The behavior cloning loss is then defined as the discrepancy between the actions generated by the current policy and those in the updated

control sequence:

$$\mathcal{L}_{\text{BC}}(\boldsymbol{\theta}, \mathbf{O}, \bar{\mathbf{A}}, \mathcal{E}) := \frac{1}{2\beta} \sum_{t=0}^{T} \|\boldsymbol{\mu}(\mathbf{o}_t; \boldsymbol{\theta}) + \boldsymbol{\sigma}(\mathbf{o}_t; \boldsymbol{\theta}) \odot \boldsymbol{\epsilon}_t - \bar{\mathbf{a}}_t\|_2^2, \tag{8}$$

where $\mathcal{E} = \{\boldsymbol{\epsilon}_0, \boldsymbol{\epsilon}_1, \ldots \boldsymbol{\epsilon}_T\}$ is same injected noises use to initialize open-loop sequence $\mathbf{A}$.

**Theorem 1.** *The decoupled trajectory gradient in* (4) *equals the negative gradient of the behavior cloning loss in Equation* (8), *i.e.,* $\tilde{\nabla}_{\boldsymbol{\theta}} \mathcal{J}(\mathbf{s}_0, \boldsymbol{\theta}, \mathcal{E}) = -\nabla_{\boldsymbol{\theta}} \mathcal{L}_{BC}(\boldsymbol{\theta}, \mathbf{O}, \bar{\mathbf{A}}, \mathcal{E})$.

*Proof.* Given that the observation sequence $\mathbf{O}$ is generated by rolling out the policy, we have $\mathbf{a}_t = \boldsymbol{\mu}(\mathbf{o}_t; \boldsymbol{\theta}) + \boldsymbol{\sigma}(\mathbf{o}_t; \boldsymbol{\theta}) \odot \boldsymbol{\epsilon}_t$. Therefore, the gradient of the behavior cloning loss simplifies to

$$\nabla_{\boldsymbol{\theta}} \mathcal{L}_{\text{BC}}(\boldsymbol{\theta}, \mathbf{O}, \bar{\mathbf{A}}, \mathcal{E}) = \sum_{t=0}^{T} \frac{1}{\beta} (\mathbf{a}_t - \bar{\mathbf{a}}_t) \frac{\partial \mathbf{a}_t}{\partial \boldsymbol{\theta}}. \tag{9}$$

Substituting E.q. (7) into 9 resulting in

$$\nabla_{\boldsymbol{\theta}} \mathcal{L}_{\text{BC}}(\boldsymbol{\theta}, \mathbf{O}, \bar{\mathbf{A}}, \mathcal{E}) = \sum_{t=0}^{T} \frac{1}{\beta} (-\beta \nabla_{\mathbf{a}_t} \mathcal{J}) \frac{\partial \mathbf{a}_t}{\partial \boldsymbol{\theta}} = -\sum_{t=0}^{T} (\nabla_{\mathbf{a}_t} \mathcal{J}) \frac{\partial \mathbf{a}_t}{\partial \boldsymbol{\theta}}. \tag{10}$$

Finally, substituting $\nabla_{\mathbf{a}_t} \mathcal{J}$ from E.q. (6) into (9) and rearranging terms completes the proof. $\square$

Theorem 1 highlights the close connection between feedback policy optimization and open-loop trajectory optimization. Iteratively applying gradient ascent with decoupled policy gradient (4) can be interpreted as alternating between two stages: (1) generating trajectories by rolling out the current policy and improving them through trajectory optimization, and (2) distilling the optimized trajectories back into the policy.

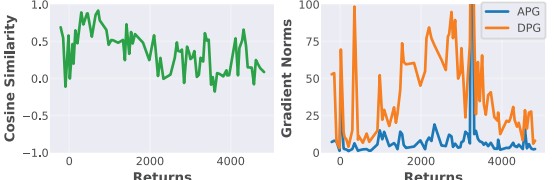

**Control regularization** We now discuss the dropped bias (5), which involves adding multiple terms of the form $\left( \frac{\partial r_t}{\partial \mathbf{a}_t} \frac{\partial \mathbf{a}_t}{\partial \mathbf{o}_t} \frac{d\mathbf{o}_t}{d\mathbf{s}_t} \frac{d\mathbf{s}_t}{d\boldsymbol{\theta}} \right)$. Here, $\frac{\partial r_t}{\partial \mathbf{a}_t}$ captures how the reward at timestep $t$ changes with the action $\mathbf{a}_t$, while the term $\left( \frac{\partial \mathbf{a}_t}{\partial \mathbf{o}_t} \frac{d\mathbf{o}_t}{d\mathbf{s}_t} \frac{d\mathbf{s}_t}{d\boldsymbol{\theta}} \right)$ reflects how past trajectory influence current decision making, i.e., $\mathbf{a}_t$. Altogether, the bias term (5) quantifies how the previous experiences influence current actions via coupling through the shared policy, and thereby impacts long-term return. In contrast, the decoupled policy gradient (4) captures how the current action affects future states, but ignores the interdependence between actions.

Figure 3: **Comparison of APG** $\nabla_{\boldsymbol{\theta}} \mathcal{V}$ **and DPG** $\tilde{\nabla}_{\boldsymbol{\theta}} \mathcal{V}$ **on Hopper with full state observation.** Both gradients are computed from the same set of $\boldsymbol{\theta}$ values collected during SHAC training. The x-axis shows the return for each $\boldsymbol{\theta}$. **Left:** Cosine similarity is generally positive, indicating $\tilde{\nabla}_{\boldsymbol{\theta}} \mathcal{V}$ is a valid ascent direction. **Right:** Gradient norm between APG and DPG. In this experiment, conducted in state space where $g = \texttt{identity}$, the control regularization term $\mathcal{B}$ acts as a residual connection. As a result, APG generally exhibits a smaller norm compared to DPG.

Figure 3 compares the APG $\nabla_{\boldsymbol{\theta}} \mathcal{V}$ and DPG $\tilde{\nabla}_{\boldsymbol{\theta}} \mathcal{V}$, both computed with respect to the short-horizon actor loss (3). The cosine similarity between the two gradients is positive in most cases, indicating that $\tilde{\nabla}_{\boldsymbol{\theta}} \mathcal{V}$ generally provides a valid ascent direction for policy improvement. Another noteworthy observation is the difference in norm between the full analytical policy gradient $\nabla_{\boldsymbol{\theta}} \mathcal{V}$ and our quasi-policy-gradient estimate, DPG $\tilde{\nabla}_{\boldsymbol{\theta}} \mathcal{V}$. When we conduct the experiment on state space, i.e., $\mathbf{o}_t = \texttt{identity}(\mathbf{s}_t) = \mathbf{s}_t$, the additive control regularization (5) operates acts similarly to a residual connection within the computation graph. The additive residual connection contribute to a smoother optimization landscape, making the norm of full APG generally smaller than DPG, as illustrate on Figure 3. This is not the case when the policy is conditioned on high-dimensional visual observations. As we will show shortly in Section 4, when the sensor model $g$ is involving complex rendering process, adding the regularization term $\mathcal{B}$ tends to increase the overall gradient norm, which hinder optimization.

**Experimental validation** A comparison between full APG and our DPG in full-state space is provided in Appendix D.1, while results under visual observations are presented in Section 4.

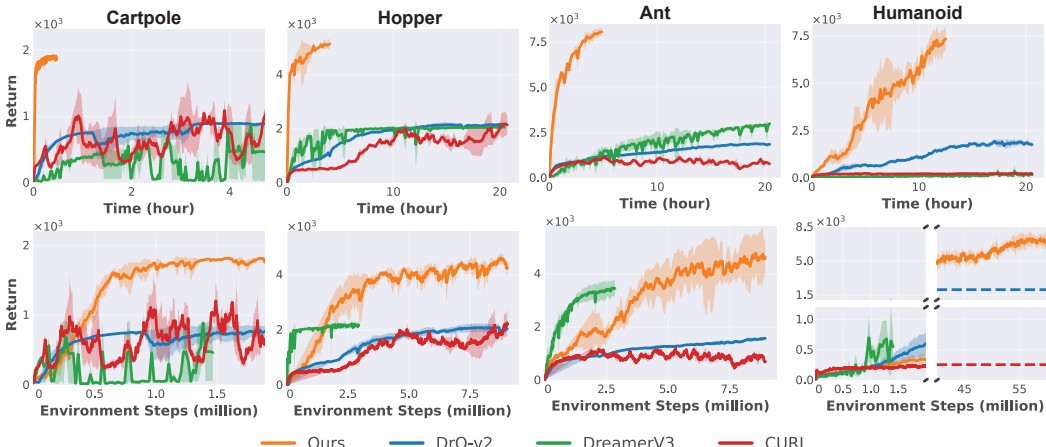

Figure 4: **Comparison with RL: our method achieves substantial speedups and significantly higher rewards across all tasks.** Each curve shows the average performance over five random seeds, with shaded areas indicating standard deviation. In the humanoid task, dashed lines represent the final rewards attained by each algorithm at the end of training. The top row highlights wall-clock efficiency; the bottom row illustrates sample efficiency, with curves truncated at the maximum number of simulation steps for better visualization.

## 3.2 Decoupled visual based analytical policy gradient

Here, we introduce Decoupled Visual-Based Analytical Policy Gradient (D.Va), a visual policy learning method built upon the decoupled policy gradient formulation. Our method is an on-policy algorithm that updates the policy using parallel simulation to generate short-horizon trajectories. Following SHAC, rollouts resume from previous endpoints and reset at task termination. Trajectories are discarded after each iteration to reduce I/O overhead. To capture temporal cues such as velocity and acceleration, we stack three consecutive image frames—following common practice in prior work [Hafner et al., 2023, Mu et al., 2025]. We also provide an ablation study on the number of stacked frames in Appendix D.3. The stacked frames are then encoded by a convolutional network to produce a latent representation $\mathbf{h}_t$ for the actor. The critic $V_\phi$ in Equation (3) plays a crucial role in achieving good overall performance. (See Appendix D.2) For efficiency, we train the critic in the low-dimensional state space $\mathcal{S}$ as opposed to the observation space $\mathcal{O}$. Although the critic is trained using state information, the policy remains state-agnostic throughout the entire training process, enabling direct deployment to downstream tasks without requiring access to privileged state information. Full algorithm details are provided in Appendix B.1.

## 4 Experiment

We design our experiments to compare the proposed method against common visual policy learning algorithms on GPU-accelerated simulation. Performance is evaluated based on final return, wall-clock time, and the number of environment steps. All hyperparameters are listed in Appendix C, while additional details on setup are provided in the Appendix E.

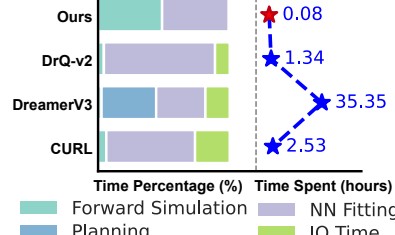

Figure 5: **Training time for Ant over 1M steps.** Left: phase percentages, where "planning" in DreamerV3 refers to rollouts by learned world model. Right: absolute times used per 1M steps. Most time in visual RL is spent fitting neural networks.

### 4.1 Comparison to RL methods

**Baseline** We compare our method with: (1) DrQ-v2 [Yarats et al., 2021], a model-free method combining image augmentations with DDPG [Lillicrap et al., 2015]; (2) CURL [Laskin et al., 2020a], a model-free RL approach using contrastive learning and SAC [Haarnoja et al., 2018]; and (3) DreamerV3 [Hafner et al., 2023], a model-based algorithm that learns a world model and uses it for planning. All RL baselines are implemented with parallelized simulation to take advantage of faster forward rollout.

**Results** Our approach achieves comparable sample efficiency to existing methods; however, it excels in wall-clock time and final returns, as shown in Figure 4. The discrepancy between sample and wall-clock time efficiency is because RL methods reuse past experiences through replay buffers, whereas our on-policy method discards samples after each iteration. Consequently, RL methods spend

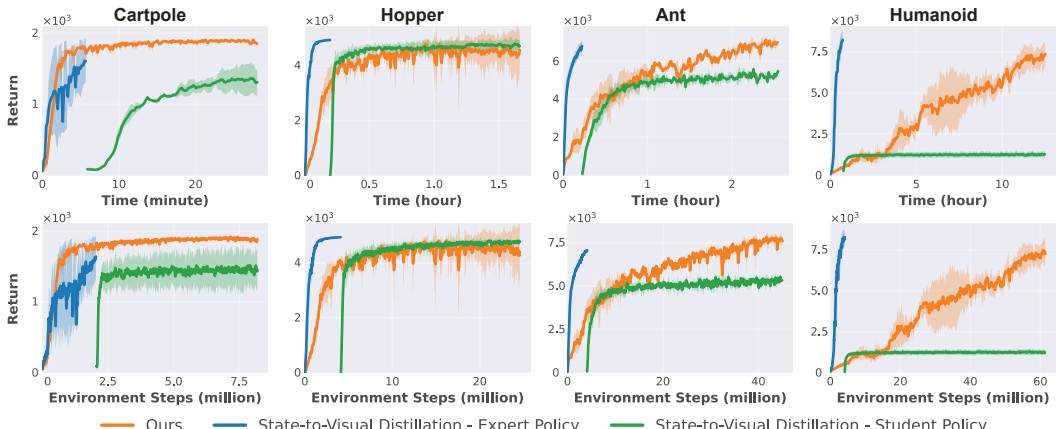

Figure 6: **Compare with State-to-Visual distillation:** Our method matches distilled policy in final return and computation time on three simple tasks, but significantly outperforms it on the more challenging Humanoid locomotion task. The student policy's x-axis is shifted to account for expert training time. Our implementation builds on Mu et al. [2025], with an enhanced expert training phase by replacing SAC with SHAC.

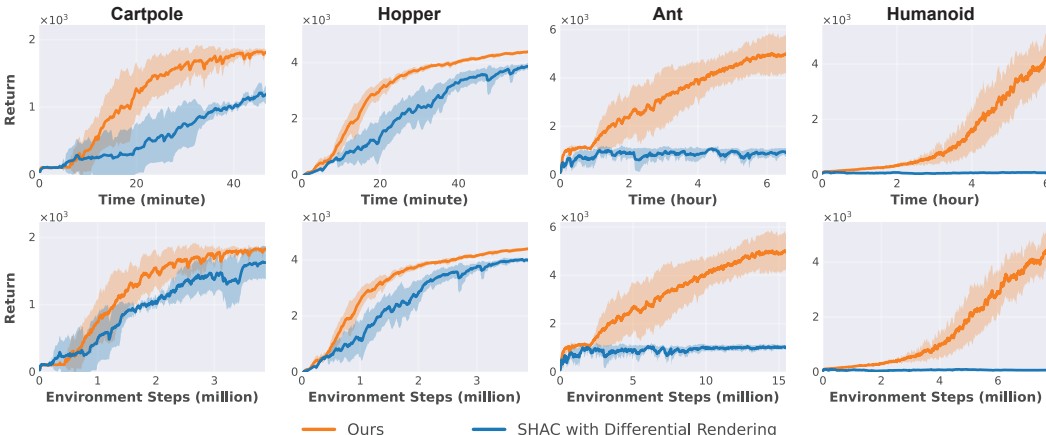

Figure 7: **Compare to SHAC with differential Rendering:** Our method matches SHAC on 2D tasks (Cartpole and Hopper) and outperforms SHAC on more challenging 3D tasks (Ant and Humanoid).

more time updating neural networks—such as fitting Q-functions in DrQ-v2—than collecting new data. As illustrated in Figure 5, only a negligible proportion of time is spent on forward simulation for the RL baselines, indicating limited potential for speedup from faster simulators. In contrast, our method allocates a comparable amount of time to both forward simulation and backward policy updates, suggesting it could further benefit from better simulation.

## 4.2 Comparison to method using privileged simulation

**State-to-visual distillation**    Another class of popular methods [Loquercio et al., 2021, Chen et al., 2023] training visual policies by first learning an expert policy with privileged state access, then transferring knowledge to a visual policy via DAgger [Ross et al., 2011]. Among these, Mu et al. [2025] proposes two key design choices—early stopping when the behavior cloning loss is low and using off-policy data from a replay buffer, which greatly reduces computation and improves performance. Our implementation of State-to-Visual distillation is based on the approach proposed by Mu et al. [2025], with one key modification: we use SHAC in place of SAC for expert training. We observe that SHAC consistently outperforms model-free SAC in settings where differentiable simulation is available Xu et al. [2021], resulting in further reduced computational time and improved final returns for training State-to-Visual distillation.

**Results**    Our method performs comparably to state-to-visual distillation on three relatively easy tasks in terms of both computation time and final rewards. In these tasks, both approaches achieve returns on par with the expert policy trained in the state space. However, on the more challenging

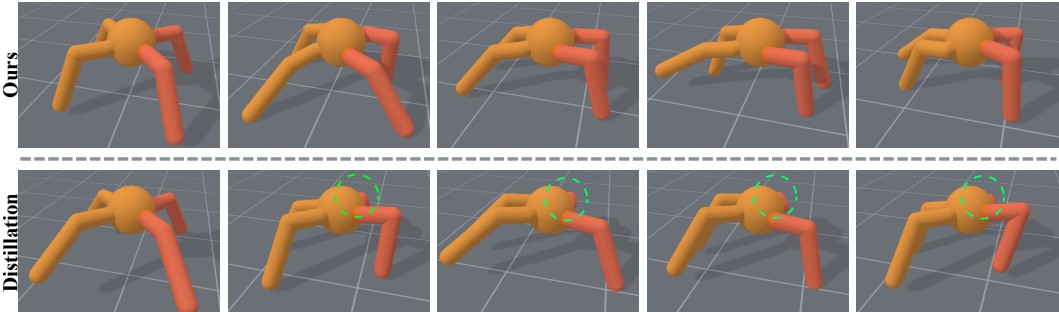

Figure 8: Our method learns policies that are camera-aware, whereas the distilled policy from expert often adopts postures that are partially occluded or blocked from the camera view.

humanoid running task, our method significantly outperforms state-to-visual distillation—achieving high final returns, while distilled policy plateaus at substantially lower values.

**Key difference**   Here, we highlight the key difference between our method and state-to-vision distillation. While both can be viewed as behavior cloning from another policy, the source of the mimicked policy differs fundamentally. In state-to-visual distillation, actions are imitated from a frozen "expert" policy. In contrast, our method mimics actions from the "teacher" that provides incremental improvements to the current policy. We argue that learning from an incremental "teacher" may be more effective than learning from a fixed "expert", especially for complex tasks. First, expert actions may differ significantly from those produced by the current policy, making them harder to imitate accurately. Second, expert policy may provide ineffective corrective feedback in those state spaces that are visited by the current policy but rarely seen during its own training phase. In other words, the expert cannot handle situations for which it has no prior experience. We hypothesize that these two factors contribute to the performance gap observed on the humanoid task.

Another subtle but noteworthy difference is in the postures generated by the learned policies. As illustrated in Figure 8, we empirically find that our method tends to produce more camera-aware behaviors, whereas the distilled policy often results in self-occluded postures. Although both the "expert" policy used in distillation and the "teacher" corrections in our method are agnostic to camera views when providing supervision, the on-policy and iterative nature of our approach may lead to important differences. During the student policy distillation phase, imitating actions from unblocked visual inputs may be easier and converge quicker than learning from occluded views, which can introduce ambiguity. Since our method continually discards outdated rollouts and relies on recent data, the training process may be implicitly biased toward favoring trajectories that offer clearer, more informative perspectives.

## 4.3   Comparison to SHAC with differentiable rendering

Finally, we compare our method against analytical policy gradient approaches that incorporate differentiable rendering. To the best of our knowledge, no existing open-source baseline combines differentiable rendering with analytical policy gradient method; therefore, we provide our own implementation.

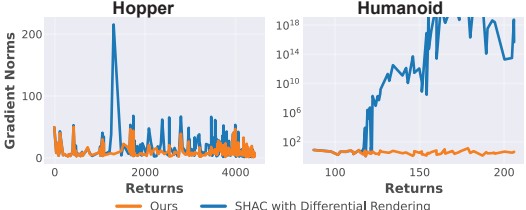

We implement differentiable rendering using PyTorch3D [Ravi et al., 2020], in contrast to prior works [Wiedemann et al., 2023, Liu et al., 2024], which rely on a learned renderer. Using a differentiable renderer based on computer graphics eliminates the need to optimize a separate neural network and helps avoid compounding errors from distribution shifts as the scene evolves.

Figure 9: **Comparison of gradient norms between SHAC with differentiable rendering and D.VA**: In this experiment, conducted in the visual space where sensor model $g$ represents a complex rendering process, the control regularization term $\mathcal{B}$ adds to a noisy optimization landscape. As a result, SHAC generally exhibits a larger gradient norm compared to D.VA.

Further details on our parallel rendering setup are provided in Appendix B.2. We then train the visual policy end-to-end under the SHAC framework. For a fair comparison, we use identical simulations and neural architectures for both methods, and the value functions are all defined on a low-dimensional state space.

**Results** Figure 7 compares our method with SHAC across four benchmark problems. We find the performance of SHAC highly dependent on whether the task is 2D or 3D. In the 2D tasks, i.e., Cartpole and Hopper, SHAC achieves performance similar to ours. However, in 3D tasks, our method consistently outperforms SHAC, with SHAC failing to learn effective locomotion. We hypothesize that the discrepancy arises from the noisy optimization landscape introduced by the complex rendering process. In contrast to the low-dimensional state space—where the control regularization term $\mathcal{B}$ (5) acts as a residual connection and the norm of the full APG is generally smaller than that of DPG (see Figure3)—the high-dimensional visual space involves a more complex sensor model $g$, which includes 3D transformations and a rasterization process. As a result, the additive control regularization term contributes to a noisier optimization landscape. As shown in Figure 9, the gradient norm of SHAC with differentiable rendering is generally larger than that of our method. Notably, for 3D tasks, the gradient norm in SHAC can rapidly exceed $10^{15}$, making the backward signal pure noise. In addition to smoother optimization, several other factors make our method preferable to SHAC for training visual policy in practice. First, our method is significantly more memory efficient, as shown in Figure 10.

Second, as scene complexity increases, i.e., the number of meshes and the number of vertices per mesh, SHAC's memory usage grows rapidly. This is because the Jacobian with respect to each mesh vertex must be stored to construct the computation graph. In contrast,

Table 1: Backward time for a single training episode.

|  | | Hopper | Ant | Humanoid |
|---|---|---|---|---|
| Ours | 0.11(s) | 0.19(s) | 0.21(s) | 0.68(s) |
| SHAC | 0.36(s) | 0.51(s) | 0.58(s) | 1.38(s) |

our method avoids storing these large Jacobians, resulting in a relatively stable memory footprint. Therefore, our method is more suitable for tasks involving a greater number of objects and higher-resolution meshes. Third, our method reduces the computational overhead during neural network updates by avoiding the multiplication of large Jacobian matrices associated with rendering during the backward pass. As shown in Table 1, our backward simulation is 2–3× faster than SHAC, measured on the same machine. Lastly, developing high-quality differentiable rendering software demands substantial engineering effort. In contrast, our method does not depend on such software, enabling easier integration into existing simulation ecosystems [Todorov et al., 2012, Xian et al., 2024], with the potential to handle multi-modal observation such as point clouds or LiDAR scans.

## 5 Related Work

**Visual policy learning** Techniques integrating learned image encoders [Finn et al., 2015, Mnih et al., 2015] have shown promising results in visual policy learning. Building on these approaches, algorithms such as Kostrikov et al. [2020], Yarats et al. [2021] improve performance through data augmentation, Laskin et al. [2020a,b], Stooke et al. [2021] leverage contrastive learning, and Hafner et al. [2019, 2023], Hansen et al. [2022] address the visual-control problem by learning world models for online planning. While these methods effectively tackle image-control challenges, they often require extensive environmental interaction and suffer from computational inefficiencies. In this work, we introduce a method that leverages differentiable simulation to reduce training time.

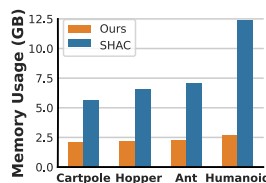

Figure 10: **Peak memory usage during training. Our method is 3–5× more memory efficient than SHAC.** Each task is run with 32 parallel environments and a short horizon length of 32.

**Policy learning with differentiable simulation** Analytical policy gradient methods [Freeman et al., 2021, Qiao et al., 2021, Mora et al., 2021] have gained traction with the rise of differentiable simulation. Among them, SHAC [Xu et al., 2021] mitigate noisy optimization via short-horizon rollouts and a value function, making it a core technique in many downstream robotics applications [Schwarke et al., 2024, Song et al., 2024]. In a subsequent work, Xing et al. [2025] further enhanced performance by adding an entropy term to the optimization objective, smoothing the optimization landscape and improving stability.

Recent efforts have adapted APG to visual policy learning: Wiedemann et al. [2023] incorporates differentiable rendering, while Heeg et al. [2024] uses cropped vision features for quadrotor racing. Luo et al. [2024] propose a hierarchical design separating control and perception. However, these approaches remain task-specific and engineering-heavy. In contrast, our method generalizes across tasks and achieves higher computational efficiency than prior visual policy learning methods.

# 6    Limitation and Future

A key limitation of our method is that it requires both the system dynamics and the reward function to be differentiable. Differentiable dynamics often rely on smoothing or approximations, which can reduce modeling accuracy, while designing differentiable reward functions is itself nontrivial. An important direction for future work is to relax these requirements by integrating alternative optimization techniques, such as random search [Howell et al., 2022, Liu et al., 2025b,a] or finite-difference methods [Mania et al., 2018].

# 7    Conclusion

Our algorithm, D.Va, is a computationally efficient method for training visual policies utilizing differentiable simulation and first-order policy gradients. With this approach, we are able to train complex visual policies within hours using modest computational resources. We believe that this improvement in computational efficiency can unlock new possibilities for the robotics community, enabling practical end-to-end training of policies from raw observations. However, this limitation is shared by all baseline methods, as training directly in the real world is often impractical. Future research should focus on how to effectively transfer the success of training visual policies in simulation to real-world scenarios.

## Acknowledgments and Disclosure of Funding

This work is supported by the National Science Foundation under award NSF FRR 2238066. Any opinions, findings, and conclusions or recommendations expressed in this material are those of the authors and do not necessarily reflect the views of the National Science Foundation.

We additionally thank the reviewers for their valuable suggestions and Davis Zong for assisting with additional experiments during the paper revision.

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

# Appendix

## A  Additional derivation

### A.1  Derivative of analytical policy gradient separation

In this section, we provide a detailed derivation of how the analytical policy gradient $\nabla_{\boldsymbol{\theta}}\mathcal{V}$ (1) can be decomposed into the decoupled policy gradient $\tilde{\nabla}_{\boldsymbol{\theta}}\mathcal{V}$ (4) and control regularized term $\mathcal{B}$ (5). We achieve this decomposition through pattern matching.

We begin with the decoupled policy gradient (4), which is obtained by taking the expectation over the gradient of individual trajectories. The gradient of each trajectory, in turn, is computed by summing the gradients of the temporal running rewards, as follows:

$$\tilde{\nabla}_{\boldsymbol{\theta}}\mathcal{J}(\mathbf{s}_0,\boldsymbol{\theta},\mathcal{E}) = \sum_{t=0}^{T}\tilde{\nabla}_{\boldsymbol{\theta}}r_t. \tag{11}$$

The gradient of the running reward, $\tilde{\nabla}_{\boldsymbol{\theta}}r_t$, is composed of two partial derivatives. First, the immediate reward $r_t$ is directly influenced by the action taken at time step $t$, denoted $\mathbf{a}_t$. This yields the partial derivative term:

$$\frac{\partial r_t}{\partial \mathbf{a}_t}\frac{\partial \mathbf{a}_t}{\partial \boldsymbol{\theta}}.$$

Second, the reward $r_t$ also depends on the state $\mathbf{s}t$, which itself depends on the previous state $\mathbf{s}t-1$ and action $\mathbf{a}_{t-1}$. In the decoupled formulation, this dependency propagates backward through time, leading to a recursive gradient computation. Specifically, the second term is

$$\frac{\partial r_t}{\partial \mathbf{s}_t}\frac{d\tilde{\mathbf{s}}_t}{d\boldsymbol{\theta}},$$

where the derivative $\frac{d\tilde{\mathbf{s}}_t}{d\boldsymbol{\theta}}$ is given recursively by: $\frac{d\tilde{\mathbf{s}}_t}{d\boldsymbol{\theta}} = \frac{\partial \mathbf{s}_t}{\partial \mathbf{s}_{t-1}}\frac{d\tilde{\mathbf{s}}_{t-1}}{d\boldsymbol{\theta}} + \frac{\partial \mathbf{s}_t}{\partial \mathbf{a}_{t-1}}\frac{\partial \mathbf{a}_{t-1}}{\partial \boldsymbol{\theta}}$. Altogether, we have

$$\tilde{\nabla}_{\theta}r_t = \frac{\partial r_t}{\partial \mathbf{a}_t}\frac{\partial \mathbf{a}_t}{\partial \boldsymbol{\theta}} + \frac{\partial r_t}{\partial \mathbf{s}_t}\frac{d\tilde{\mathbf{s}}_t}{d\boldsymbol{\theta}}. \tag{12}$$

Figure 11 illustrates a typical backward flow on decoupled policy gradient starting with the reward $r_2$.

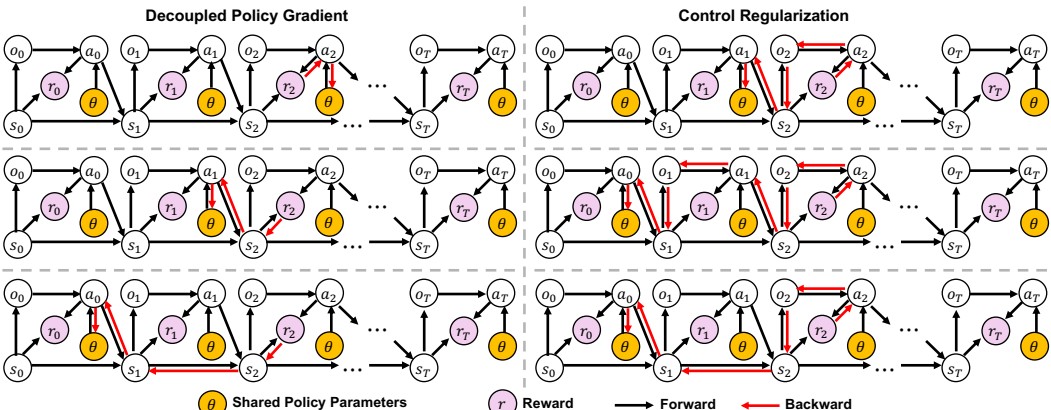

Figure 11: **Backward gradient start with $r_2$.** Left: the backward gradient flow on the decoupled policy gradient. Right: the backward flow on control regularization. Each subplot in the row starts from the reward node $r_2$ and traces backward along the computation graph until it reaches the policy parameters $\boldsymbol{\theta}$, illustrating a partial contribution to the total derivative.

We now examine how the control regularization term (5) is formulated. Following the structure illustrated in Figure 11, the control regularization is constructed by summing a series of terms of the form:

$$\frac{\partial r_t}{\partial \mathbf{a}_t}\frac{\partial \mathbf{a}_t}{\partial \mathbf{o}_t}\frac{d\mathbf{o}_t}{d\mathbf{s}_t}\frac{d\mathbf{s}_t}{d\boldsymbol{\theta}}.$$

Here, the total derivative $\frac{d\mathbf{s}_t}{d\boldsymbol{\theta}}$ can be computed recursively as follows:

$$\frac{d\mathbf{s}_t}{d\boldsymbol{\theta}} = \frac{\partial \mathbf{s}_t}{\partial \mathbf{a}_{-1}}\Big(\frac{\partial \mathbf{a}_{-1}}{\partial\boldsymbol{\theta}} + \frac{\partial \mathbf{a}_{-1}}{\partial \mathbf{o}_{t-1}}\frac{d\mathbf{o_{t-1}}}{d\mathbf{s}_{t-1}}\frac{d\mathbf{s}_{t-1}}{d\boldsymbol{\theta}}\Big) + \frac{\partial \mathbf{s}_t}{\partial \mathbf{s}_{t-1}}\frac{d\mathbf{s}_{t-1}}{d\boldsymbol{\theta}}. \tag{13}$$

Altogether, the control regularization term is given in Eq. (5). Thus, the full analytical policy gradient is the sum of the decoupled policy gradient and the control regularization term:

$$\nabla_{\boldsymbol{\theta}}\mathcal{V} = \tilde{\nabla}_{\boldsymbol{\theta}}\mathcal{V} + \mathcal{B}. \tag{14}$$

# B  Additional implementation details

## B.1  D.VA algorithm

**Critic Learning**  Our critic training follows SHAC, minimizing the mean squared error over collected trajectories:

$$\mathcal{L}_\phi = \mathbb{E}_{\mathbf{s}\in\{\tau^{(i)}\}}\Big[\|V_\phi(s) - \tilde{V}(\mathbf{s})\|_2^2\Big], \tag{15}$$

where

$$\tilde{V}(\mathbf{s}_t) = (1-\lambda)\Big(\sum_{k=1}^{h-t-1}\lambda^{k-1}G_t^k\Big) + \lambda^{h-t-1}G_t^{h-t}, \tag{16}$$

is the estimated value function and is treated as a constant target during critic learning. Here, $G_t^k = \Big(\sum_{l=0}^{k-1}\gamma^l R(\mathbf{s}_{t+l}, \mathbf{a}_{t+l})\Big) + \gamma^k V_{\phi'}(\mathbf{s}_{t+k})$ represents the $k$-step return from time $t$, and $V_{\phi'}$ is a delayed critic function used to stabilize the training process [Mnih et al., 2015].

**Full algorithm**  Below, we summarize our Decoupled Visual-Based Analytical Policy Gradient (D.VA) algorithm.

---

**Algorithm 1:** D.VA (Decoupled Visual Based Analytical Policy Gradient)

---
**Function** `Rollout()`
  Initialize $N$ initial states $\mathbf{s}_0$.
  **for** $t = 0$ to $h-1$ **do**
    with `torch.no_grad()`:
      compute pixel images $\mathbf{o}_t = g(\mathbf{s}_t)$.
    Sample actions $a_t \sim \pi_{\boldsymbol{\theta}}(\mathbf{o}_t)$, simulate and compute rewards $r_t$ and next states $\mathbf{s}_{t+1}$.
  **end for**
  Collect $N$ trajectories $\tau = \{(\mathbf{s}_t, a_t, r_t)\}_{t=0}^{h-1}$ and compute actor loss $\mathcal{L}_{\boldsymbol{\theta}}$ via Eq. (3).
  **Return** $\tau, \mathcal{L}_{\boldsymbol{\theta}}$
**Function** `Main()`
  Initialize $\pi_{\boldsymbol{\theta}}, V_\phi, V_{\phi'} \leftarrow V_\phi$
  **for** $epoch = 1$ to $M$ **do**
    Generate $N$ short-horizon trajectories $\tau$ and compute actor loss $\mathcal{L}_{\boldsymbol{\theta}}$ by calling `Rollout()`.
    Compute decoupled policy gradient $\tilde{\nabla}_{\boldsymbol{\theta}}\mathcal{V}$ and update $\pi_{\boldsymbol{\theta}}$ with Adam.
    Fit value function $V_\phi$ via critic loss (15) and update delayed target $V_{\phi'} \leftarrow \alpha V_{\phi'} + (1-\alpha)V_\phi$.
  **end for**

---

## B.2  Differentiable Rendering

In this section, we present the construction of our differentiable renderer, designed to facilitate end-to-end training.

Given a state vector $\mathbf{s} = [\mathbf{q}^\top, \dot{\mathbf{q}}^\top]^\top$, where $\mathbf{q} \in \mathbb{R}^n$ represents joint positions and $\dot{\mathbf{q}}$ denotes joint velocities, which have dimension $m-1$ or $m$ depending on the presence of quaternions. We compute $m$ homogeneous transformations $\mathbf{T}$ for forward kinematics. These transformations encode the rotation and translation from the world frame to each joint frame and are used to transform geometry meshes defined in local joint frames to assemble the full robot mesh. Lighting and camera poses are also updated using these transformations to set up the full scene. Finally, we render the scene using PyTorch3D via rasterization. Figure 12 summarizes the whole process.

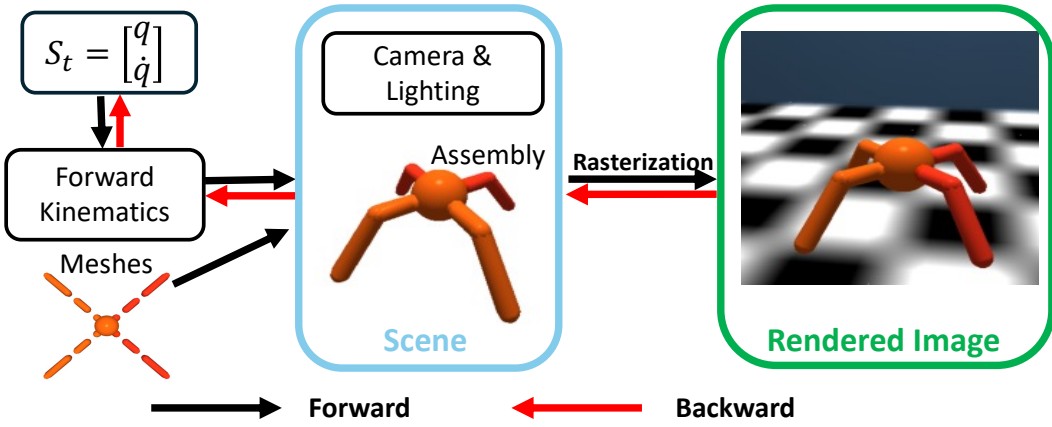

Figure 12: Diagram for differentiable rendering

# C Hyperparameters

In this section, we describe our hyperparameter tuning process and share additional insights gained during experimentation. Detailed hyperparameter values are provided at the end.

### C.1 How we tune hyperparameters

For all baseline methods, we initialized hyperparameters using values reported in the original papers. We then identified the most sensitive hyperparameters and tuned them sequentially. Additional attention is given to those hyperparameters emphasized in the original works. This process involved over 500 experiments, with some trials manually terminated early when it became evident that the chosen hyperparameters were suboptimal. In contrast, we did not apply additional tuning to our proposed method, as the combination of hyperparameters from state-based SHAC and the encoder architecture from DrQV2 already yielded strong performance.

We summarize our key findings from the tuning process and present the final hyperparameter values used in the following sections.

### C.2 Key findings and experimental settings

**DrQ-v2** We implement DrQ-v2 with parallelized forward simulation to improve wall-clock time efficiency. During each training episode, we concurrently collect new samples using the current policy across multiple parallel environments. After data collection, we perform several update steps on the Q-functions using the data from the replay buffer. The forward pass accounts for only a small portion of the total training time with parallelization, as shown in Figure 5. The majority of computation time is instead dedicated to updating the agent's neural network, i.e., Q-function and Actor. We found that the ratio between agent update steps and new sample collection plays a critical role in achieving good performance. While performing multiple agent updates per environment step can improve sample efficiency, it may negatively impact wall-clock time performance. In addition, we observe that excessive updates can harm final policy performance due to outdated data from the replay buffer. Therefore, we carefully tune both the number of parallel environments and the number of updates per training episode, aiming to maximize wall-clock efficiency and final return. The hyperparameters used in our experiments are listed in Table 2. The primary difference from the original DrQ-v2 setup is that we increase both the number of parallel environments and the number of agent updates per step, while the ratio is kept the same. Unless otherwise noted, the same parameters are used across all four tasks. We kept neural architecture identical to that used in the original paper and summarize in Table 3 and 4:

**CURL** We also apply parallelized forward simulation to CURL to accelerate training. Similar to DrQ-v2, the update frequency plays a critical role in achieving strong performance. We tune the number of parallel environments to optimize wall-clock efficiency within the available GPU memory budget. The training parameters used for CURL are listed in Table 5. Notably, CURL uses a similar

Table 2: DrQ-v2 training parameters

| Parameter name | Value |
| --- | --- |
| Number of parallel environments | 32; 16 for Cartpole |
| Number of agent updates | 16; 8 for Cartpole |
| Replay buffer capacity | $10^6$ |
| Action repeat | 2 |
| Mini-batch size | 256 |
| N-step returns | 3 |
| Discount factor $\gamma$ | 0.99 |
| Learning rate | $10^{-4}$; ($8 \times 10^{-5}$ for Humanoid) |
| Critic Q-function soft-update rate | 0.01 |
| Exploration stddev. clip | 0.3 |
| Exploration stddev. schedule | `linear`(1.0, 0.1, 500000); `linear`(1.0, 0.1, 2000000) for humanoid |

Table 3: DrQV2 encoder architectures

| Parameter name | Value |
| --- | --- |
| Input image size (height $\times$ width) | $84 \times 84$ |
| Convolution kernel size | 3, 3, 3, 3 |
| Convolution output channel size | 32, 32, 32, 32 |
| Convolution activation function | Relu |

Table 4: DrQ-v2 actor-critic architecture

| Task name | Trunk size | Policy network | Critic network |
| --- | --- | --- | --- |
| Cartpole | 50 | [1024, 1024] | [1024, 1024] |
| Hopper | 50 | [1024, 1024] | [1024, 1024] |
| Ant | 50 | [1024, 1024] | [1024, 1024] |
| Humanoid | 100 | [1024, 1024] | [1024, 1024] |

architecture to DrQ-v2 (see Table 3, and 4); both are adopted from Yarats et al. [2020]. To ensure a fair comparison, we use the same architecture for both algorithms, avoiding confounding factors introduced by architectural differences.

**DreamerV3** Our implementation of DreamerV3 builds upon the open-source repository available at `https://github.com/NM512/dreamerv3-torch`. We parallelized the environment stepping in our implementation; however, we observed that this parallelization has minimal impact on performance—consistent with the findings reported by the author of the repository. Additionally, DreamerV3 is a memory-intensive algorithm, which limits the degree of parallelism we can apply. The detailed hyperparameters for training are mostly kept the same as those used in Hafner et al. [2023] and listed in Table 6. The neural architecture remains unchanged from the Hafner et al. [2023].

**D.Va (Ours)** Architecture details: Stacked images are first processed by an encoder to generate a hidden state. The encoder is a 4-layer convolutional network, identical to that used in DrQ-v2 (Table 3). The hidden state is then passed to the actor network to generate actions. The actor network consists of a trunk network and a policy network, following the design of DrQ-v2. The trunk network is a single linear layer followed by layer normalization. The policy and critic network is adopted from the state-based SHAC: MLP network with ELU activation and layer normalization. The detailed network architectures are provided in Table 7.

Table 5: CURL training parameters

| Parameter name | Value |
|---|---|
| Number of parallel environments | 16 |
| Number of agent updates | 8 |
| Replay buffer capacity | $10^5$ |
| Action repeat | 2 |
| Batch size | 32 |
| Discount factor $\gamma$ | 0.99 |
| Actor learning rate | $10^{-3}$ |
| Critic learning rate | $10^{-3}$ |
| Adam $(\beta_1, \beta_2)$ for actor and critic | (0.9, 0.99) |
| Q-function soft-update rate | 0.01 |
| Initial temperature | 0.1 |
| Temperature learning rate | $10^{-4}$ |
| Adam $(\beta_1, \beta_2)$ for temperature | (0.5, 0.99) |

Table 6: DreamerV3 training parameters

| Parameter name | Value |
|---|---|
| Image size (height $\times$ width) | ($64 \times 64$) |
| Number of parallel environments | 4 |
| Batch size | 16 |
| Batch length | 64 |
| Train ratio | 512 |
| Action repeat | 2 |
| Replay buffer capacity | $10^6$ |
| Action repeat | 2 |
| Discount factor $\gamma$ | 0.997 |
| Discount lambda $\lambda$ | 0.95 |
| Actor learning rate | $3 \times 10^{-5}$ |
| Critic learning rate | $3 \times 10^{-5}$ |
| Actor-critic adam epsilon | $10^{-5}$ |
| World model learning rate | $10^{-4}$ |
| World model adam epsilon | $10^{-8}$ |
| Critic EMA decay | 0.98 |
| Reconstruction loss scale | 1.0 |
| Dynamics loss scale | 0.5 |
| Representation loss scale | 0.1 |
| Actor entropy scale | $3 \times 10^{-4}$ |
| Return normalization decay | 0.99 |

Table 7: D.Va actor-critic architecture

| Task name | Trunk size | Policy network | Critic network |
|---|---|---|---|
| Cartpole | 64 | [64, 64] | [64, 64] |
| Hopper | 128 | [128, 64, 32] | [64, 64] |
| Ant | 128 | [128, 64, 32] | [64, 64] |
| Humanoid | 256 | [256, 128] | [128, 128] |

For training, we apply a linear decay schedule to adjust the learning rate over episodes, with specific hyperparameters provided in Table 8.

Table 8: D.Va training parameters

| Parameter name | Cartpole | Hopper | Ant | Humanoid |
|---|---|---|---|---|
| Short horizon length $h$ | 32 | | | |
| Number of parallel environments $N$ | 64 | | | |
| Actor learning rate | 0.002 | | | |
| Critic learning rate | 0.0002 | | 0.002 | 0.0005 |
| Target value network $\alpha$ | 0.2 | | | 0.995 |
| Discount factor $\gamma$ | 0.99 | | | |
| Value estimation $\lambda$ | 0.95 | | | |
| Adam $(\beta_1, \beta_2)$ | $(0.7, 0.95)$ | | | |
| Number of critic training iterations | 16 | | | |
| Number of critic training minibatches | 4 | | | |

**State-to-visual Distillation**  The architecture is kept identical to that used for D.Va, which can be found in Table 3 and 7. As described earlier, the architecture is constructed by simply concatenating the DrQv2 encoder with the SHAC state-based architecture, ideally, no method is unfairly favored. The hyperparameters are tuned following the guidelines of Mu et al. [2025]. We found that, to make State-to-Visual Distillation work effectively, the most critical factor is the data collection strategy and the frequency of network updates, consistent with the findings reported in Mu et al. [2025]. Specifically, we collect data in a SHAC-style manner: instead of executing long, continuous trajectories, we roll out short-horizon segments that resume from the endpoint of the previous rollout. Notably, this implementation is the same as that used in Mu et al. [2025]. The detail parameters are listed in Table 9

Table 9: State-to-visual Distillation training parameters

| Parameter name | Value |
|---|---|
| Short horizon length $h$ | 32 |
| Number of parallel environments $N$ | 64 |
| Learning rate | 0.002 |
| Adam $(\beta_1, \beta_2)$ | $(0.7, 0.95)$ |
| Batch size | 128 |
| Early stop threshold | 0.1 |
| Maximum reply buffer size | $10^5$ |

**SHAC with differentiable rendering**  The neural architecture is kept identical to ours and is detailed in Table 3 and Table 7. For 3D tasks such as Ant and Humanoid, we observed that the gradient norm quickly diverges to infinity as the horizon length $h$ increases. To address this, we use an even smaller horizon length compared to the one used for training state-based SHAC. The number of parallel environments is also reduced due to GPU memory constraints. The specific parameter values are provided in Table 10.

## D   Additional Experiment

### D.1   Compare DPG to SHAC on state space

We provide additional experiments to valid using decoupled policy gradient is enough to gain good performance in many scenarios. Here, we conduct on state space, with a single line change on the original SHAC code: `actions = actor(obs)` to `actions = actor(obs.detach())`. We kept all hyperparameters identical to those reported in SHAC. Figure 13 compares the training performance of the SHAC with full analytical policy gradient 1 to our decoupled policy gradient 4. We find that

Table 10: Visual-SHAC training parameters

| Parameter name | Cartpole | Hopper | Ant | Humanoid |
|---|---|---|---|---|
| Short horizon length $h$ | 32 | 32 | 8 | 32 |
| Number of parallel environments $N$ | 64 | 64 | 32 | 32 |
| Critic learning rate | 0.0002 | | 0.002 | 0.0005 |
| Target value network $\alpha$ | 0.2 | | | 0.995 |
| Actor learning rate | 0.002 | | | |
| Discount factor $\gamma$ | 0.99 | | | |
| Value estimation $\lambda$ | 0.95 | | | |
| Adam $(\beta_1, \beta_2)$ | $(0.7, 0.95)$ | | | |
| Number of critic training iterations | 16 | | | |
| Number of critic training minibatches | 4 | | | |

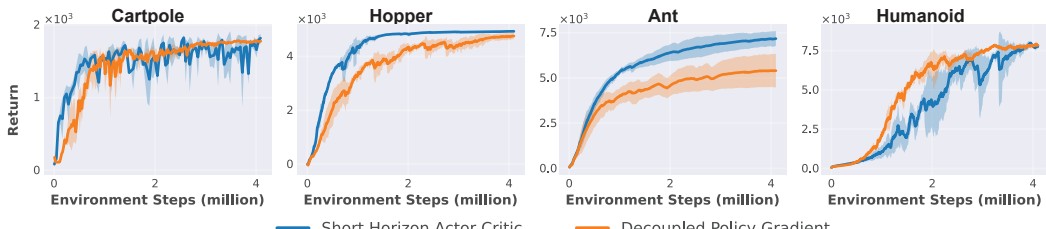

Figure 13: **Comparison between DPG and SHAC on state space:** Experiments are conducted with low-dimensional state observations and results are averaged over five random seeds. All hyperparameters are kept identical to those used in the original SHAC. DPG still achieves comparable results in the settings that favor SHAC.

the decoupled policy gradient achieves performance comparable to SHAC on low-dimensional state spaces, even under settings that favor SHAC.

## D.2 Ablation on Value Function

In this section, we present an ablation study on the value function. As shown in Figure 14, without the value function, our method fails to learn an effective policy. This result highlights the critical role of the value function in analytical policy gradient methods.

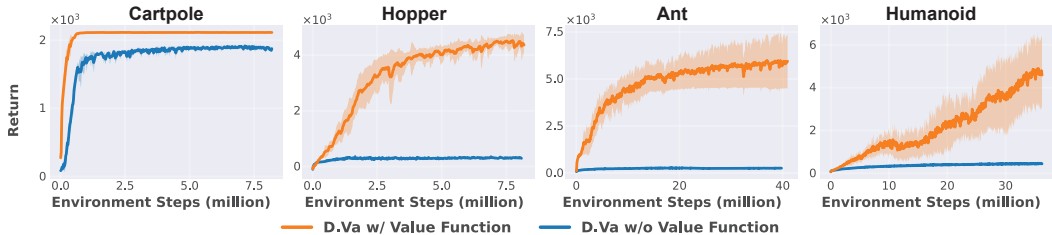

Figure 14: Ablation study on value function. D.Va without value function fail to learn effective control for most of the tasks.

## D.3 Ablation on number of frames

In this section, we present an ablation study on the effect of the number of concatenated frames used as input to the policy. Table 11 shows the final return achieved with different numbers of stacked frames as input to the policy. We observe that as long as the number of frames is not extremely low (e.g., one), the method achieves comparable final performance.

Table 11: Final Return Achieved with Different Number of Stacked Frames

| Number of frames | Cartpole | Hopper | Ant | Humanoid |
|---|---|---|---|---|
| 1 | $2068 \pm 29.83$ | $4398.48 \pm 266.80$ | $5681 \pm 1344.52$ | $5596.94 \pm 937.92$ |
| 2 | $2115 \pm 21.20$ | $5055.23 \pm 6.73$ | $7418 \pm 862.39$ | $5342.00 \pm 1680.70$ |
| 3 | $2139 \pm 22.42$ | $5067.13 \pm 18.14$ | $7218 \pm 986.00$ | $7475.77 \pm 812.48$ |
| 4 | $2155 \pm 25.49$ | $5055.03 \pm 2.08$ | $9680 \pm 222.27$ | $7498.79 \pm 623.11$ |
| 5 | $2095 \pm 22.5$ | $5072.33 \pm 14.62$ | $8286 \pm 880.50$ | $6345.61 \pm 37.76$ |
| 6 | $2140 \pm 29.84$ | $5077.63 \pm 23.74$ | $8243 \pm 799.45$ | $6578.59 \pm 717.24$ |

## D.4 Additional tasks

We include an additional quadruped walking task using ANYmal [Hutter et al., 2016], with third-party side-view images as input. As shown in Figure 15, our method learns a visual locomotion policy within 20 minutes on a single GPU.

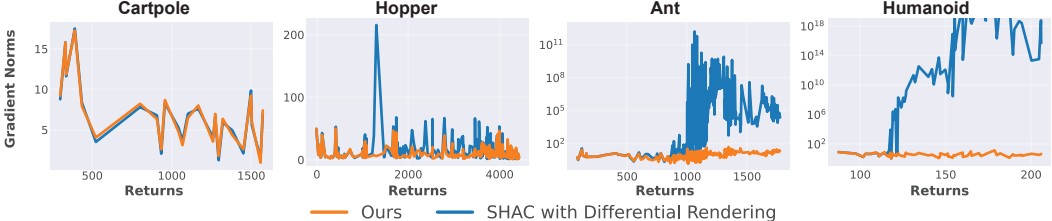

Figure 15: ANYmal locomotion: Our method is able to learn ANYmal locomotion from purely third-party visual input within 20 minutes.

## D.5 More gradient norm analysis

Figure 16 shows the gradient norms of the visual policy computed using SHAC and D.Va, respectively. We observe that the gradient norms for the two 3D tasks blow up, which explains why SHAC with differentiable rendering fails to learn an effective visual policy for these tasks, as shown in Figure 7.

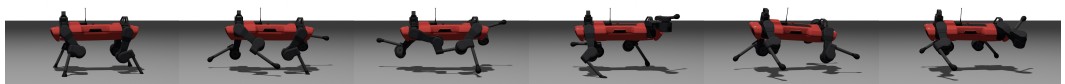

Figure 16: Gradient norms computed using SHAC and D.Va are shown for all environments. The gradient norms for the Ant and Humanoid tasks blow up.

# E Setup

## E.1 Tasks Descriptions

We select four classical RL tasks across different complexity levels. The camera views are similar to those used in Yarats et al. [2021], as illustrated in Figure 17. Except for Cartpole, where the camera is fixed to the world frame, all other cameras track the position of the robot's base joint. The motion of Cartpole and Hopper is constrained to 2D, whereas Ant and Humanoid are free to move in 3D space. The body of the Ant may block the view of some of its legs due to the side-view camera setup, making the environment partially observable. In contrast, the joints in all other environments remain visible from the camera regardless of posture, resulting in fully observable settings.

The reward functions are identical to those used in the SHAC paper, except for the Cartpole system, where we add a health score to prevent the cart from moving off-screen. The details are summarized as follows:

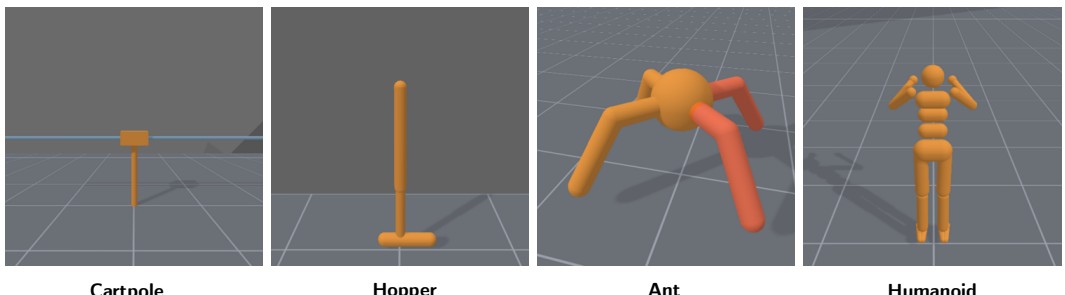

Figure 17: Camera views of each environment in ManiskillV3

**Cartpole:** The running rewards are defined as

$$R := 10 - \theta^2 - 0.1\dot{\theta}^2 - 0.05x^2 - 0.1\dot{x}^2, \tag{17}$$

where $\theta$, $x$ denote the angle of pole from upright position and position of cart; and $\dot{\theta}$, $\dot{x}$ are the angular and linear velocity respectively. The total trajectory length is 240, and early termination is triggered when Cartpole is outside the camera views, i.e., $|x| \geq 2.5$.

**Hopper:** The running rewards are defined as

$$R := v_x - (\frac{\theta}{30°})^2 + R_{\text{height}} - 0.1\|\mathbf{a}\|, \tag{18}$$

where $v_x$ is the forward velocity, $\theta$ is the orientation of base joint and

$$R_{\text{height}} = \begin{cases} -200\Delta_h^2, \ \Delta_h \leq 0 \\ \Delta_h, \ \Delta > 0 \end{cases} ; \ \Delta_h = \text{clip}(h + 0.3, -1, 0.3), \tag{19}$$

is designed to penalize the low height state. The total trajectory length is 1000, and early termination is triggered when the height of the hopper is lower than -0.45m.

**Ant:** The running rewards are defined as

$$R := v_x + 0.1R_{\text{up}} + R_{\text{heading}} + p_z - 0.27, \tag{20}$$

where $v_x$ is forward velocity, $R_{\text{up}}$, $R_{\text{heading}}$ is the projection of base orientation in upright and forward direction, encouraging the agent to be vertically stable and run straightforward, $p_z$ is the height of the base. The total trajectory length is 1000, and early termination is triggered when the height of the ant is lower than 0.27m.

**Humanoid:** The running rewards are defined as

$$R := v_x + 0.1R_{\text{up}} + R_{\text{heading}} + R_{\text{height}} - 0.002\|\mathbf{a}\|, \tag{21}$$

where $v_x$ is forward velocity, $R_{\text{up}}$, $R_{\text{heading}}$ is the projection of base orientation in the upright and forward direction, and

$$R_{\text{height}} = \begin{cases} -200\Delta_h^2, \ \Delta_h \leq 0 \\ \Delta_h, \ \Delta > 0 \end{cases} ; \ \Delta_h = \text{clip}(h - 0.84, -1, 0.1). \tag{22}$$

The total trajectory length is 1000, and early termination is triggered when the height of the torso is lower than 0.74m.

### E.2 Simulation

We use the same differentiable simulation framework proposed in SHAC as the underlying dynamics model. For the three benchmark RL methods and the state-to-visual tasks, we employ ManiSkill-V3 [Tao et al., 2024] for rendering. In contrast, for visual-SHAC, we implement a custom differentiable rendering pipeline using PyTorch3D, as detailed in Appendix B.2. All software components are GPU-accelerated and parallelized. We evaluate our method under both rendering pipelines. To ensure a fair comparison, all experiments presented in Section 4 are conducted using

the same rendering setup across different methods. We find the ManiSkill implementation to be fairly efficient—approximately $3\times$ faster than our differentiable rendering pipeline—and therefore, we use differentiable rendering only when necessary. However, aside from the difference in forward rendering speed, we find that the final return and sample efficiency of our method remain similar across both rendering pipelines.

## E.3 Hardware

All experiments are conducted on a single NVIDIA GeForce RTX 4080 GPU (16GB) with an Intel Xeon W5-2445 CPU and 256GB RAM. Unlike the case of simulating dynamics alone—where tens of thousands of environments can be parallelized at once—heterogeneous rendering requires significantly more memory. As a result, our hyperparameter tuning is carefully constrained to stay within the available memory budget.

