# OpenReview forum: "Accelerating Visual-Policy Learning through Parallel Differentiable Simulation"
_NeurIPS.cc/2025/Conference — NeurIPS 2025 spotlight_

### Official Review · Reviewer_YuQp · 2025-06-03

**Clarity:** 3
**Significance:** 2
**Originality:** 3
**Rating:** 5
**Confidence:** 4

**Summary:**

The paper studies the problem of optimizing visual policies using first-order gradients with differentiable simulators. One major problem is that computing first-order gradients in high-dimensional image spaces is computationally expensive. To address this, the paper proposes decoupled policy gradient (DPG) that avoids back-propagation through the rendering process. The paper proves that DPG is identical to open-loop trajectory optimization. Finally, the paper integrates the proposed method with SHAC, resulting in decoupled visual-based analytic policy gradient (D.Va). The final proposed algorithm achieves superior performance compared to model-free and model-based RL algorithms, including DrQ-v2, DreamerV3, and SHAC with differential rendering.

**Questions:**

- Could you also compare your method with online RL algorithms that leverage privileged information, such as [1]?
- Is there a particular reason you chose not to provide privileged information to the critic?
- Why did you select DreamerV3 rather than TD-MPC2 [2] as the model-based RL baseline? Since your experiments use the ManiSkill-V3 simulator, where TD-MPC2 has already been tested, TD-MPC2 would be the more appropriate choice.

[1] Edward S. Hu, et al., Privileged Sensing Scaffolds Reinforcement Learning, ICLR 2024 \
[2] Nicklas Hansen, et al., TD-MPC2: Scalable, Robust World Models for Continuous Control, ICLR 2024

**Ethical Concerns:**

["NO or VERY MINOR ethics concerns only"]

**Final Justification:**

The proposed method is simple yet effective, and the presentation is clear. While it would be better to include more baselines that use privileged information, the current results are already sufficiently strong. All of my concerns have been addressed. Accordingly, I am raising my score to 5.

**Limitations:**

The authors have adequately addressed the limitations of the proposed method in the conclusion section.

**Quality:**

2

**Strengths And Weaknesses:**

Strengths
- (Originality) Extension of first-order gradient policy optimization from low-dimensional state spaces to high-dimensional image spaces is novel.
- (Quality) The proposed method is simple yet effective.
- (Clarity) The gradient analysis in Figure 3 is very helpful for understanding the effectiveness of the proposed method.
- (Quality) The paper compares the proposed method with many different baselines, including zeroth-order RL algorithms and first-order policy gradient algorithms.

Weaknesses
- (Quality) The authors should have fairly compared their proposed method with RL with privileged information.
- (Significance) I am not sure that the proposed method is better than first training an expert policy using state information only (which is 5~10x more efficient as shown in Figure 13) and then distilling it to an image-based policy (which is also efficient as shown in Figure 6).

---

> ### Author Rebuttal · Authors · 2025-07-28
>
> # Author’s response to reviewer YuQp
>
> We sincerely thank reviewer YuQp for their valuable feedback on our paper and their positive comments on the novelty and overall good performance of our method. We address the reviewer’s concerns as follows:
>
> **Q1: I am not sure that the proposed method is better than first training an expert policy using state information only (which is 5~10x more efficient as shown in Figure 13) and then distilling it to an image-based policy (which is also efficient as shown in Figure 6).**
>
> > **A1**: We argue that the wall-clock time for student-teacher distillation methods should be measured end-to-end—that is, from training the state-based teacher to obtaining the final visual policy. Our method eliminates the two-stage process and trains the visual policy directly from scratch. This one-stage training offers a key advantage: learning begins immediately, whereas in student-teacher pipelines, visual policy training only starts after the teacher is fully trained. In our experiments, our method often reaches near-final performance before distillation in the two-stage baseline even begins.
>
> > In addition to training time, another important aspect is the final returns that can be achieved. As shown in Figure 6, our method continues to improve in the visual humanoid locomotion task, ultimately achieving returns comparable to expert policies trained with privileged state input. In contrast, the two-stage methods plateau much earlier, with significantly lower final performance.
>
> **Q2:  Is there a particular reason you chose not to provide privileged information to the critic? The authors should have fairly compared their proposed method with RL with privileged information.**
>
> > **A2**: We do use privileged state information to train the critic, as noted in Lines 172–174 and 254–256. Additionally, for a fair comparison, all baselines in Sections 4.2 (state-to-visual distillation) and 4.3 (SHAC + differentiable rendering) also incorporate privileged state information during training.
>
> **Q3: Could you also compare your method with online RL algorithms that leverage privileged information, such as [1]?**
>
> >**A3**: Thank you for pointing us to [1]. While [1] focuses on more challenging POMDP settings, our experiments are conducted on standard visual RL benchmarks that are closer to fully observable environments (see Figure 14). In this case, adding additional sensor modalities may not lead to significant performance improvements.
>
> > However, we especially appreciate the reviewer to introduce [1] to us and the associated S3 benchmark —it looks like a strong testbed for evaluating methods like ours under partial observability. This also aligns with Reviewer GuqR’s feedback and highlights an exciting direction for future work. Based on [1]’s taxonomy, our method could be seen as using both a privileged critic and a privileged world model (via differentiable simulation), and we’re looking forward to exploring this further.
>
> **Q4: Why did you select DreamerV3 rather than TD-MPC2 [2] as the model-based RL baseline? Since your experiments use the ManiSkill-V3 simulator, where TD-MPC2 has already been tested, TD MPC2 would be the more appropriate choice.**
>
> >**A4**: We initially considered both TD-MPC2 and DreamerV3 as baselines. However, in our experiments on visual MuJoCo benchmarks—especially the more challenging humanoid tasks—we observed that TD-MPC2 failed to learn effective locomotion policies. In contrast, DreamerV3 performed more robustly and achieved significantly better results across visual tasks.
>
> > Additionally, as noted in the original TD-MPC2 paper, their method is not specifically designed for visual RL tasks. Their reported results only include simple 2D visual task environments and do not report more complex visual locomotion tasks as evaluated in our paper. Based on both empirical performance and intended design scope, we chose DreamerV3 as a more suitable and competitive model-based RL baseline for our visual control experiments.
>
> > ### Table: Final Returns Achieved for Humanoid Tasks with Visual Inputs
>  | Method       | Final Return    |
> |--------------|----------------------------|
> | Ours         | 7475.77 ± 812.48           |
> | Dreamer-V3   | 1517.00 ± 571.63           |
> | TD-MPC2      | 50.70 ± 10.52              |
>
> **References:**
>
> [1] Edward S. Hu, et al., “Privileged Sensing Scaffolds Reinforcement Learning”, ICLR 2024
>
> [2] Nicklas Hansen, et al., “TD-MPC2: Scalable, Robust World Models for Continuous Control”, ICLR 2024

---

> ### Comment · Reviewer_YuQp · 2025-08-01
>
> Thank you for your detailed response. I realize now that I misread Figure 6. It might be clearer to relabel the curves as
> "state policy" (instead of "expert policy") and "visual policy" (instead of "student policy"), using the same color for both but different line styles (solid vs dashed)d. That said, this is only a minor suggestion—please feel free to keep the figure as is if you prefer. In addition, my earlier question about the critic was off the mark. I simply wanted to understand why you chose not to use both the state and the observation together. All my concerns have been fully addresses. Thank you.

---

> > ### Author Response · Authors · 2025-08-01
> >
> > Thank you for your suggestion. We will modified the label in Figure 6 for clarification in our final version. Also, thank you for introducing valuable related work and benchmarks.

---

### Official Review · Reviewer_cpR5 · 2025-07-02

**Clarity:** 3
**Significance:** 3
**Originality:** 3
**Rating:** 5
**Confidence:** 3

**Summary:**

This paper presents D.VA, a novel on-policy AC algorithm for visual policy learning in differentiable simulation. It proposes to decouple the visual part gradients and builds a two-stage optimization method, which can be interpreted as an open-loop trajectory optimization and distillation. The method is evaluated on four classic RL environments, demonstrating better wall-clock time and even outperforming off-policy algorithms in sample efficiency. The authors also provide the profiling results and a comparison with teacher-student distillation with state-based teacher policy, which is very informative.

**Questions:**

- Can you explain in details why the incremental teacher is more easily to mimic than the expert frozen teacher in lines 222 to 224?
- Based on the two-step optimization approach, it is similar to the idea of DIPO and DDiffPG that first optimize the action respect to the Q-gradient and then imitate the optimized action. Therefore, can author explain the connection with those off-policy algorithms and is it possible to extend the current method into an off-policy fashion?
- In Fig. 3, the grad norm of DPG is much larger than APG, however, the authors claim that the training of DPG is actually more stable in line 39. Is there evidence to support this claim as I didn't find anything in the method or experiments?

**Ethical Concerns:**

["NO or VERY MINOR ethics concerns only"]

**Final Justification:**

The authors provided a comprehensive evaluation to support claims made in the manuscript. In the rebuttal, the authors further discussed the potential future directions and limitations in details, making the paper more solid.

**Limitations:**

Limitation is discussed in the conclusion.

**Quality:**

3

**Strengths And Weaknesses:**

Strength:
- the idea is novel and well-motivated
- the paper is generally well-written and easy to follow
- provides comprehensive and diverse experiments to give better understanding of the method

Weakness:
- the authors claim that the decoupled policy gradients provides smooth optimization and stable training. However, there is no evidence to support and in Fig. 3 the gradient norm is even larger than the counterpart, which I think larger gradient usually leads to unstable learning.
- the tasks are relatively simple. In line 137 the authors mentioned the impact of dropped bias term, which captures the interdependence between the actions. Therefore it is interesting to explore how the proposed method works in more challenging tasks, such as long-horizon tasks.

---

> ### Author Rebuttal · Authors · 2025-07-28
>
> # Author’s response to reviewer cpR5
>
> We sincerely thank reviewer cpR5 for their valuable feedback on our paper and their positive comments on the novelty of our method and the comprehensiveness of our experiments.We address the reviewer’s concerns as follows:
>
> **Q1: the authors claim that the decoupled policy gradients provides smooth optimization and stable training. However, there is no evidence to support and in Fig. 3 the gradient norm is even larger than the counterpart, which I think larger gradient usually leads to unstable learning.
> In Fig. 3, the grad norm of DPG is much larger than APG, however, the authors claim that the training of DPG is actually more stable in line 39. Is there evidence to support this claim as I didn't find anything in the method or experiments?**
>
> > **A1**: Thank you for the question. Figure 9 provides empirical support that our decoupled policy gradient improves training stability—showing much smaller gradient norms than SHAC. In high-dimensional visual tasks SHAC gradients can explode (up to $10^{15}$), stalling learning. In contrast, our gradients stay well-behaved and enable smoother optimization.
>
> > As for Figure 3, it's a low-dimensional proof-of-concept (as noted in Lines 157–162) meant to validate that our surrogate provides a descent direction—not to claim improved stability. While the gradient norm there is higher than SHAC, both are within a manageable range (below $10^2$), and standard practices like gradient clipping are effective, as supported by our results (Appendix Figure 13) and prior work [1, 2].
>
> **Q2: The tasks are relatively simple.**
>
> > **A2**: We evaluate our method across a range of classical RL environments with varying levels of difficulty. Notably, on the challenging visual humanoid locomotion task, our approach achieves up to 4× higher return compared to baselines (see Figure 4,6,7), while also demonstrating the fastest wall-clock training time.
>
> > In addition, we are working on adding experiments on more complex robotics-oriented tasks, such as the Anymal robot running task, which features more complex dynamics and partial observability due to large body occlusions. Our method is able to learn effective locomotion policies on Anymal in under 15 minutes, further highlighting its efficiency and adaptability.
>
> **Q3: Can you explain in details why the incremental teacher is more easily to mimic than the expert frozen teacher in lines 222 to 224?**
>
> > **A3**: Thank you for the question. There are two main reasons why an incremental teacher (online) is easier to mimic than a frozen expert (offline). First, as noted in Line 224, a frozen expert can produce actions that differ significantly from the student policy. This naturally raises a crucial implementation choice: how close to mimic newly collected data. Mimicking too closely slows down training due to heavy supervised updates, while mimicking too loosely can degrade data quality generated in future cycles and detract from the final performance (as the student collects data under its own imperfect policy, the expert’s training distribution is no longer align well with the states visited by the student). Following best practices from [4], we extensively tuned crucial hyperparameters such as update ratios and added tricks like early stopping to get strong performance. In contrast, our method avoids this trade-off by directly applying gradient ascent, simplifying training and improving stability.
>
> > Second, as mentioned in Lines 225–228, a fixed expert doesn’t adapt to state distribution visited by students. The incremental teacher, however, co-evolves with the student, providing more on-policy guidance and resulting in a higher final performance gain.
>
> **Q4: In line 137 the authors mentioned the impact of dropped bias term, which captures the interdependence between the actions. Therefore it is interesting to explore how the proposed method works in more challenging tasks, such as long-horizon tasks.**
>
> > **A4**: Thank you for highlighting this point. We agree that it is an interesting and important direction to explore how our method performs in long-horizon settings. However, our current approach is built on top of SHAC, which already truncates the planning horizon to a short window. This makes it difficult to isolate the effect of dropping the bias term in the context of long-horizon tasks. Meanwhile, extending our method within a hierarchical RL framework—where a high-level planner provides guidance in subgoal, may help address long-horizon dependencies, as demonstrated in prior work [3]. We view this as a promising direction for future research.
>
> **Q5: Based on the two-step optimization approach, it is similar to the idea of DIPO and DDiffPG that first optimize the action respect to the Q-gradient and then imitate the optimized action. Therefore, can author explain the connection with those off-policy algorithms and is it possible to extend the current method into an off-policy fashion?**
>
> > **A5**: Thank you for pointing us to DIPO and DDiffPG. We were not aware of these works during the course of our project, and we find them highly relevant and inspiring to us. Below we address the reviewer’s questions based on our understanding of these two works:
>
> > **1. Connection to off-policy methods like DIPO and DDiffPG**:
> Our method is fundamentally on-policy, where state-action pairs are discarded after each training episode. In that sense, our approach is conceptually closer to DDiffPG than DIPO. While we present our method as a two-step optimization (trajectory optimization followed by supervised policy update) for the sake of conceptually understanding decoupled policy gradient, in practice we directly compute the decoupled policy gradient and apply gradient ascent rather than explicitly generating new trajectories and behavior cloning as done in DIPO/DDiffPG. This design removes the need to manually tune the learning rate for the trajectory optimization step, which can be sensitive. As noted in Equation (7), our method supports arbitrarily small learning rates in the trajectory optimization phrase, allowing it to handle highly non-smooth trajectories. The difficulty of optimizing such nonsmooth trajectories is absorbed into the behavioral cloning phrase (Equation 8), which is generally easier to optimize in practice. This unified treatment automatically balances optimization difficulty across the two stages.
>
> > **2. Extending to off-policy learning**:
> Yes, our approach can potentially be extended to an off-policy setting. We are actively exploring this in our concurrent work. A natural way to do so is to divide decoupled-policy gradient into the two stages—trajectory optimization and behaviour—and apply multiple updates to each phrase using existing data. However, we find there are many challenges in this setting, such as choosing a suitable learning rate for trajectory optimization on non-smooth objectives and maintaining a suitable ratio between new data collection and policy updates. Extending our method to off-policy learning is definitely important direction and we are currently investigating.
>
> **References**:
>
> [1] Cheng, An-Chieh, et al. “Navila: Legged robot vision-language-action model for navigation.”, arXiv preprint arXiv:2412.04453 (2024).
>
> [2] Hyung Ju Suh, et al. “Do differentiable simulators give better policy gradients?”, ICML, 2022.
>
> [3] Cheng, An-Chieh, et al. "Navila: Legged robot vision-language-action model for navigation.", arXiv preprint arXiv:2412.04453 ,2024.
>
> [4] Tongzhou Mu, et al.. “When should we prefer state-to-visual dagger over visual reinforcement learning?”, AAAI, 2025.

---

> > ### Author Response · Authors · 2025-08-07
> >
> > Thank you for your efforts in helping us improve our work. Please let us know if you have any further concerns that we can address.

---

### Official Review · Reviewer_vvNV · 2025-07-03

**Clarity:** 3
**Significance:** 3
**Originality:** 3
**Rating:** 5
**Confidence:** 3

**Summary:**

The authors propose to study mechanisms for efficient policy learning in visual tasks. They take advantage of differentiable simulators to compute an approximation for the full gradient that that not differentiate through rendering terms. They find this is a strong speed-up over baselines.

**Questions:**

Given how fast this trains, why not run more experiments on more environments? Normally I wouldn't ask such a question but these seem like fairly light compute environments and it would allow you to actually make the claims you do in the conclusion like ""
- Do the results in Figure 9 hold in the other 2 environment used on the paper?
- The claim on line 172 that the value function is critical for performance. Where is the experiment supporting that claim?
- I understand that there are analyses in this, but methods-wise, what is different here than just running SHAC and not differentiating through the observation function? If it's just that, should it really have such a distinct name?
- GIven how large the gradient norms are, why is gradient clipping not needed here? Or is it used?

**Ethical Concerns:**

["NO or VERY MINOR ethics concerns only"]

**Final Justification:**

My questions primarily concerned some lack of justification for some of the claims and an insufficient discussion of the limitations. The authors have addressed them and so I'm keeping my already high score.

**Limitations:**

This paper does not really address weaknesses and limitations of the method. The main limitation explicitly discussed, reliance on simulators, is not really a limitation of the method. It is hard for me to tell the authors what the limitations of their methods are, but a more substantive discussion would have improved the paper.

**Quality:**

3

**Strengths And Weaknesses:**

### Strengths
- the baselines seem to be carefully tuned and explanations are provided for the tuning procedures
- the sample complexity seems to have serious improvements over the baselines
- seeming improvements in wall-clock speed over the original method it is derived from

### Weaknesses
- addressed in questions


###  Suggestions (which you can ignore)
- This paper would be more convincing if it was run on more environments. Two of the environments, Cartpole/Hopper, are fairly straightforward to solve with any method. Seeing this work across many environments might induce people to use it.

---

> ### Author Rebuttal · Authors · 2025-07-28
>
> # Author’s response to reviewer vvNV
>
> We sincerely thank reviewer vvNV for their valuable feedback on our paper and their positive comments on the effectiveness of our method.
>
> We address the reviewer’s concerns as follows:
>
> **Q1: Given how fast this trains, why not run more experiments on more environments? This paper would be more convincing if it was run on more environments. Two of the environments, Cartpole/Hopper, are fairly straightforward to solve with any method. Seeing this work across many environments might induce people to use it.**
>
> > **A1**: We appreciate the reviewer’s suggestion to include additional environments, as broader validation is indeed valuable. In quick response, we have already added experiments on the Anymal locomotion task with side-view cameras—a more realistic robotic control scenario during the rebuttal period. Our method successfully learns effective locomotion policies in under 15 minutes.
>
> > While our initial submission focused on carefully benchmarking and tuning state-of-the-art methods—which required substantial computational resources—we recognize the importance of demonstrating versatility. After the rebuttal period, we plan to expand our experiments to a broader set of robotics-oriented tasks and include more examples in the final version.
>
> **Q2:  Do the results in Figure 9 hold in the other 2 environment used on the paper?**
>
> > **A2**: Yes, and we will include figures for other environments in our final version.
>
> **Q3 The claim on line 172 that the value function is critical for performance. Where is the experiment supporting that claim?**
>
> > **A3**: Thank you for the suggestion! We’ll include an ablation study in the final version to highlight the role of the value function. In the meantime, we’ve added a table showing the final return of DVA with and without the value function. More broadly, value functions are known to significantly aid training in analytical policy gradient methods—not only in visual tasks (see, e.g., Figure 4 in SHAC [1]).
>
> > ### Table: Final Returns Achieved With and Without Value Function
> | Environment | Without Value Function | With Value Function     |
> |-------------|------------------------|--------------------------|
> | Hopper      | 454.33 ± 49.34         | 5067.13 ± 18.14          |
> | Humanoid    | 539.56 ± 15.52         | 7475.77 ± 812.48         |
>
> **Q4: I understand that there are analyses in this, but methods-wise, what is different here than just running SHAC and not differentiating through the observation function? If it's just that, should it really have such a distinct name?**
> > **A4**: Yes, the key difference from SHAC is that we stop the gradient through the observation (rendering) function. However, we argue that this change leads to meaningful improvements—especially on visual tasks by stabilizing training (See Figure 9).
>
> > We gave it a distinct name to highlight the impact of this decoupling, similar to how methods like DrQ [2] or RLPD [3] build on existing algorithms with simple yet effective modifications. We hope that DVA can serve as a reference point for applying this strategy within analytical policy gradient methods.
>
> **Q5: Given how large the gradient norms are, why is gradient clipping not needed here? Or is it used?**
> > **A5**: Yes, we apply gradient norm clipping to both our method (DVA) and SHAC during training. However, simply clipping the norm proves insufficient when gradients become as extremely large as the $10^{15}$ values seen with SHAC in our Humanoid experiments (Figure 9). The ineffectiveness of analytical policy gradients in the presence of such extreme values has been extensively studied in previous work (e.g., [4, 5]).
>
> **Q6: This paper does not really address weaknesses and limitations of the method. The main limitation explicitly discussed, reliance on simulators, is not really a limitation of the method. It is hard for me to tell the authors what the limitations of their methods are, but a more substantive discussion would have improved the paper.**
> > **A6**: Thank you for pointing this out—we appreciate the opportunity to reflect more deeply on the limitations of our method. We agree that the reliance on simulators is a broader challenge shared by many reinforcement learning approaches, and may not uniquely characterize our method. We included it because, in practice, simulator quality often becomes a key bottleneck when transferring methods to real-world scenarios.
>
> > More specifically to our approach, one limitation is that it builds upon analytical policy gradient methods and therefore requires full differentiability of the trajectory—this includes not just the dynamics, but also the reward function. While differentiable rewards can be naturally defined for many tasks (e.g., locomotion, manipulation) it becomes more challenging for tasks with sparse or binary feedback (e.g., success/failure). Designing differentiable proxies for such rewards may require additional heuristics. We will update the paper to include a more detailed discussion of these limitations and thank the reviewer again for the helpful suggestion.
>
> **References**:
>
> [1] Jie Xu, et al., “Accelerated Policy Learning with Parallel Differentiable Simulation,” ICLR 2021.
>
> [2] Denis Yarats, et al., “Image augmentation is all you need: Regularizing deep reinforcement learning from pixels.”  ICLR. 2021.
>
> [3] Ball Philip J., et al., “Efficient online reinforcement learning with offline data.” ICML, 2023.
>
> [4] Luke Metz, et al. “Gradients are not all you need”. arXiv preprint arXiv:2111.05803, 2021.
>
> [5] Hyung Ju Suh, et al., “Do differentiable simulators give better policy gradients?” ICML. PMLR, 2022.

---

> > ### Comment · Reviewer_vvNV · 2025-08-07
> > **Feedback appreciated**
> >
> > Hi!
> > Thank you for the response. This addresses my questions but my score was mostly due to the number of environments this was tested on so I'm not going to raise it. However, my score still indicates that this paper should be accepted!

---

### Official Review · Reviewer_GuqR · 2025-07-03

**Clarity:** 3
**Significance:** 3
**Originality:** 2
**Rating:** 5
**Confidence:** 4

**Summary:**

This paper introduces D.Va, a novel method for training visual control policies efficiently with first-order analytical policy gradient using differentiable simulation.
The core innovation, the decoupling of visual observations from the computation graph during BP enables more efficient training.
The authors prove the effectiveness of their method with extensively benchmarking diverse approaches and theoretical analysis.

**Questions:**

See in weaknesses.

**Ethical Concerns:**

["NO or VERY MINOR ethics concerns only"]

**Final Justification:**

This is a technically solid paper, which could also lead to many exicting future research directions. It also provides many detailed experiment results and statistics analysis. Readers can easily follow and reproduce the results. My concern have been all well addressed after the rebuttal.

**Limitations:**

Yes.

**Paper Formatting Concerns:**

No.

**Quality:**

3

**Strengths And Weaknesses:**

## strengths
1. The decoupling visual policy learning method in the paper is novel. And the authors provide sufficient formal and experimental analysis to prove the effectiveness of the DPG(Decoupling Policy Gradient).
2. Extentive comparisons on performance, wall-clock time and memory across diverse methods are made. And the performance gains well support the method.
3. The paper is well-strucutured, and figures are clear.

## weaknesses
1. In this work, the authors use stacked frames (3 frames) to provide temporal cues. In many tasks in locomotion like running, 3 frames may be enough. How sensitive is the method to the number of frames?
2. While decoupled policy gradient has shown great performance, but as analyzed in the paper, the term of control regularization reflects how past trajectory influence current decision making. So the decoupling strategy might not be so effective on long-horizon dependent tasks.
3. Also, dropping the term of control regularization makes the policy optimization to be blind towards observation. So the reviewer suspects that the decoupling strategy might succeed easily on single task, and wonders the performance of its generalization ability.
4. $V_{\phi}$ plays a crucial role in achieving good overall performance. But actually this could only work for sim-to-real paradigm. This can also explain why the DPG is stronger than SHAC with differentiable rendering on 3D tasks, as it has access to the privileged states.

---

> ### Author Rebuttal · Authors · 2025-07-28
>
> # Author’s response to reviewer GuqR
>
> We sincerely thank reviewer GuqR for their valuable feedback on our paper and their positive comments on the novelty of the decoupling visual policy learning method.
>
> We address the reviewer’s concerns as follows:
>
> **Q1: In this work, the authors use stacked frames (3 frames) to provide temporal cues. In many tasks in locomotion like running, 3 frames may be enough. How sensitive is the method to the number of frames?**
>
> > **A1**: We conducted additional experiments to evaluate the sensitivity of our method to the number of stacked frames and report the final returns achieved below.  In short, we observe that as long as the number of frames is not extremely low (e.g., 1), the method achieves similar final performance, which corresponds to results in [1]
>
> > ### Table: Final Return Achieved with Different Number of Stacked Frames
> >
> > | Environment | Frame 1          | Frame 2           | Frame 3           | Frame 4           | Frame 5           | Frame 6           |
> > |-------------|------------------|-------------------|-------------------|-------------------|-------------------|-------------------|
> > | Hopper      | 4398.48 ± 266.80 | 5055.23 ± 6.73    | 5067.13 ± 18.14   | 5055.03 ± 2.08    | 5072.33 ± 14.62   | 5077.63 ± 23.74   |
> > | Humanoid    | 5596.94 ± 937.92 | 5342.00 ± 1680.70 | 7475.77 ± 812.48  | 7498.79 ± 623.11  | 6345.61 ± 37.76   | 6578.59 ± 717.24  |
>
>
> **Q2: The value function plays a crucial role in achieving good overall performance. But actually this could only work for sim-to-real paradigm. This can also explain why the DPG is stronger than SHAC with differentiable rendering on 3D tasks, as it has access to the privileged states.**
>
> > **A2** :Thank you for the comment. We’d like to clarify that both DVA and SHAC use privileged state information to train the value function (as noted in Lines 255–256), so the performance gap is not due to unequal access to privileged information.
>
> > The key difference lies in the optimization approach: DVA uses a surrogate decoupled gradient that leads to more stable training (which is supported by Figure 9 and discussion in Lines 266-268). As shown in Figure 9, SHAC often produces extremely large gradients (up to $10^{15}$), leading to high-variance training, while DVA maintains more smaller and regular gradient norms, leading to less variance.
>
> **Q3: While decoupled policy gradient has shown great performance, but as analyzed in the paper, the term of control regularization reflects how past trajectory influence current decision making. So the decoupling strategy might not be so effective on long-horizon dependent tasks.**
>
> > **A3**: Thank you for the insight! We agree with Reviewer GuqR that our method alone may face challenges in long-horizon planning tasks and could result in sub-optimal trajectories. However, it is unclear whether this limitation is due to the removal of the control regularization term or due to SHAC itself, which already uses a short planning horizon. Analytical policy gradient methods like ours are in general better suited for high-frequency (e.g. 50Hz) controls that involve agile motions. Many downstream applications—such as those in [2, 3, 4]—require visual feedback and quick reactions, where our method can be especially effective. To better handle long-horizon dependencies, it is still possible to integrate our approach into a hierarchical RL framework, where a high-level planner guides a low-level controller—an approach shown to be effective in prior work such as [5]. We see this as a promising direction for future research.
>
> **Q4: Also, dropping the term of control regularization makes the policy optimization to be blind towards observation. So the reviewer suspects that the decoupling strategy might succeed easily on single task, and wonders the performance of its generalization ability.**
>
> > **A4**: Thank you for raising this important point. We acknowledge that removing the control regularization term may reduce the coupling between the policy and the observation, potentially making the policy less sensitive to camera pose. We briefly mention this concern in Lines 231–233 of our paper.
>
> > However, this challenge is not unique to our method—it also arises in widely used techniques like teacher-student distillation (our second class of baselines), which are commonly applied in downstream visual-policy training for robotics tasks [1, 2, 3]. Interestingly, we observe that our method exhibits a degree of camera-awareness in practice: the learned policy often biases locomotion toward more favorable viewing angles—more so than the teacher-student distillation baselines (see Figure 8). While this is an empirical observation, it suggests that decoupling does not eliminate all sensitivity to visual input.
>
> > That said, we agree that generalization under partial observability remains a major challenge, and our current method alone may not fully address such cases. As partial observability is a long-standing issue in RL, a full treatment is beyond the current scope of this work, but we appreciate the reviewer’s feedback in helping us clarify this discussion. We see this as a promising direction for future work.
>
> **References**:
>
> [1] Shang, Wenling, et al. "Reinforcement learning with latent flow." Advances in Neural Information Processing Systems 34 (2021): 22171-22183.
>
> [2] Zhuang, Ziwen, et al. "Robot parkour learning." arXiv preprint arXiv:2309.05665 (2023).
>
> [3] Liu, Minghuan, et al. "Visual whole-body control for legged loco-manipulation." arXiv preprint arXiv:2403.16967 (2024).
>
> [4] Antonio Loquercio, et al.” Learning high-speed flight in the wild”. Science Robotics, 6(59):eabg5810, 2021.
>
> [5] Cheng An-Chieh, et al. "Navila: Legged robot vision-language-action model for navigation." arXiv preprint arXiv:2412.04453 (2024).

---

> > ### Comment · Reviewer_GuqR · 2025-08-06
> >
> > I thank the authors for detailed and careful reply.
> >
> > **Q-1**: I do enjoy seeing the table shown here, which is very informative. Thanks the authors again. If possible, I would sugges the authors including this in the paper as well.
> >
> > **Q-2,3,4**: I agree with the authors' comment.
> >
> > My concern have been all well addressed, and I will raise my rating accordingly.

---

### Decision · Program_Chairs · 2025-09-17

**Decision:**

Accept (spotlight)

**Comment:**

This paper proposes D.Va, a computationally efficient method for visual policy learning that leverages differentiable simulation by decoupling the rendering process from the computation graph. The reviewers are in unanimous agreement that this paper should be accepted (all providing a score of 5).

The proposed D.Va is novel and simple, but also fast, with significant improvements in wall-clock training time and final performance when compared to a set of strong baselines. The rebuttal effectively addressed concerns about generalization, scope of the evaluation, and stability of the gradients. Overall, this is a technically solid paper with a clear and impactful contribution to visual policy learning.